# HEAR: An EEG Foundation Model with *H*eterogeneous *E*lectrode *A*daptive *R*epresentation

## Abstract

Electroencephalography (EEG) is an essential technique for neuroscience research and brain-computer interface (BCI) applications. Recently, large-scale EEG foundation models have been developed, exhibiting robust generalization capabilities across diverse tasks and subjects. However, the heterogeneity of EEG devices not only hinders the widespread adoption of these models but also poses significant challenges to their further scaling and development. In this paper, we introduce **HEAR**, the first EEG foundation model explicitly designed to support heterogeneous EEG devices, accommodating varying electrode layouts and electrode counts. HEAR employs a learnable, coordinate-based spatial embedding to map electrodes with diverse layouts and varying counts into a unified representational space. This unified spatial representation is then processed by a novel spatially-guided transformer, which effectively captures spatiotemporal dependencies across electrodes. To support the development of HEAR, we construct a large-scale EEG dataset comprising 8,782 hours of data collected from over 150 distinct electrode layouts with up to 1,132 electrodes. Experimental results demonstrate that HEAR substantially outperforms existing EEG foundation models in supporting heterogeneous EEG devices and generalizing across diverse cognitive tasks and subjects. Our code is available at: `https://anonymous.4open.science/r/HEAR-5AAE`.

## 1 Introduction

EEG offers a dynamic and non-invasive means of monitoring brain activity by capturing electrical signals from the cerebral cortex. Its portability and real-time capabilities make it an effective tool in neuroscience research and practical BCI applications. Despite its widespread adoption, the performance of EEG decoding algorithms is fundamentally constrained by challenges, such as a low signal-to-noise ratio, substantial inter-subject variability, and task-dependent fluctuations Craik et al. (2019). Due to these challenges, traditional EEG decoding models struggle to generalize across tasks and subjects Lawhern et al. (2018), limiting their effectiveness in real-world applications.

To address these challenges, recent efforts have focused on developing EEG foundation models (FMs) Lai et al. (2025) using large-scale, unlabeled EEG datasets, which have demonstrated significantly improved generalizability across diverse tasks and subjects Jiang et al. (2024). However, despite the substantial promise of EEG FMs, their practical deployment is still hindered by the heterogeneity of EEG devices. Due to considerable differences in EEG electrode counts and layouts, scaling up EEG FMs and applying them to tasks involving novel devices with differing electrode configurations remains a significant challenge. To address this issue, current EEG FMs typically restrict input to a fixed subset of commonly shared channels, often aligned with the standard 10–20 system Wang et al. (2024a); Yang et al. (2024); Kostas et al. (2021). Although this approach facilitates dataset alignment, it substantially limits the model's ability to leverage the full spatial information inherent in EEG signals, and may result in significant performance degradation on downstream tasks.

To address this challenge, we propose Heterogeneous Electrode Adaptive Representation (HEAR), an EEG foundation model designed to support a wide range of EEG systems with varying electrode counts and layouts. HEAR can effectively model diverse electrode layouts by establishing a unified representational space for different electrode systems and developing a spatially-guided transformer

architecture. Notably, we demonstrate that HEAR can generalize across more than 150 electrode layouts and accommodate up to 1,132 electrodes. The main contributions of this work are summarized below:

- We introduce *HEAR*, the first EEG foundation model capable of accommodating heterogeneous EEG devices with varying electrode layouts and counts.
- We construct the *HEAR Dataset*, a large-scale EEG dataset comprising 8,782 hours of data across more than 150 electrode layouts, all encoded within a unified representational space.
- Extensive experimental results demonstrate that the HEAR model not only exceeds the performance of state-of-the-art (SOTA) EEG foundation models across a wide range of tasks, but also generalizes robustly to challenging testing scenarios where electrode layouts are unseen during training.

## 2 RELATED WORK

**EEG Foundation Models:** In response to the increasing demand for scalable and generalizable EEG analysis, a series of EEG FMs have recently been proposed, leveraging large-scale pretraining to enhance downstream task performance and cross-domain adaptability. Current EEG FMs can be broadly categorized into two types: temporal modeling and spatial modeling approaches. Temporal modeling approaches (such as *BENDR* Kostas et al. (2021), *Neuro-GPT* Cui et al. (2024), and *EEGPT* Wang et al. (2024a)) utilize self-attention mechanisms within the Transformer architecture to capture long-term temporal dependencies in EEG signals. In contrast, spatial modeling approaches, including *BIOT* Yang et al. (2024), *LaBraM* Jiang et al. (2024), *Brant* Zhang et al. (2023), *CBraMod* Wang et al. (2024b), and *BrainBERT* Wang et al. (2023), place additional emphasis on learning spatial representations through self-supervised learning and frequency-aware embeddings.

Table 1: Overview of strategies adopted by existing EEG FMs to address electrode heterogeneity.

| Model | Channel Input Strategy | Heterogeneity Support | |
|---|---|---|---|
| | | Variable Channels | Unseen Layouts |
| *BIOT* | Common-channel subset alignment | ✗ | ✗ |
| *BENDR* | Fixed-channel time-domain modeling | ✗ | ✗ |
| *LaBraM* | ID-based hard spatial encoding | ✓ | ✗ |
| *EEGPT* | Local electrode convolution | ✓ | ✗ |
| *BrainBERT* | Spectrogram-based encoding | ✗ | ✗ |
| *FoME* | Time-frequency fusion with attention | ✗ | ✗ |
| **HEAR** | Coordinate-based spatial embedding and modeling | ✓ | ✓ |

**Electrodes Heterogeneity Handling Strategies:** Existing architectures indicate that by jointly learning the neural representations across EEG data's spatial and temporal dimensions, models can better adapt to various downstream tasks. As summarized in Table 1, many of these EEG FMs have sought to address the challenge of heterogeneity in electrode layouts. For instance, models such as BIOT, BENDR, and LaBraM employ a channel-subset alignment strategy, whereby only the intersection set of channels across different datasets is considered. Other approaches operate at the per-channel level: EEGPT leverages local convolutions to capture spatial neighborhood information, while BrainBERT and FoME rely on time-frequency representations predicated on a fixed channel count. Notably, LaBraM incorporates a patch-based tokenizer to accommodate variable-length signals. However, its spatial encoding remains constrained by a predetermined channel count, thereby limiting its ability to generalize to unseen electrode layouts. Overall, current EEG foundation models exhibit substantial limitations when confronted with arbitrary and previously unseen electrode counts and layouts due to rigid input assumptions and insufficient spatial adaptability. This underscores the need for a more generalized framework capable of effectively handling electrode heterogeneity.

## 3 METHODOLOGY

The overall architecture of the proposed HEAR model is depicted in Figure 1. Firstly, the HEAR dataset partitions EEG recordings according to electrode layout. Each distinct electrode layout

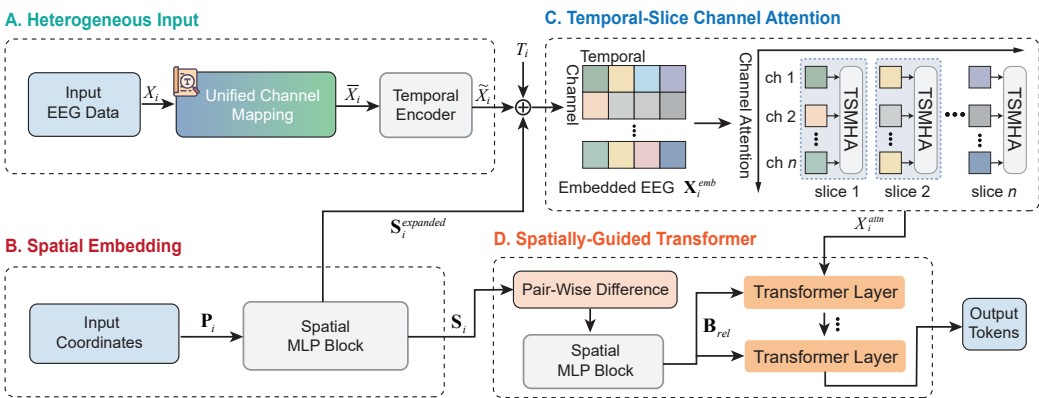

Figure 1: Overview of the HEAR model. (A) Heterogeneous Input. EEG signals are first aligned by selecting available channels, then segmented into non-overlapping temporal windows for each channel. (B) Spatial Embedding. The input coordinates are projected into a shared embedding space and injected into patch tokens to encode spatial topology. (C) Temporal-Slice Channel Attention. Attention is applied across channels within each temporal slice to capture local spatial dependencies. (D) Spatially-Guided Transformer. Pairwise spatial relationships between electrodes are used to generate attention bias, enabling layout-aware Transformer modeling.

is labeled using a unified channel dictionary, as detailed in Section 3.1. Subsequently, the input signals are projected into a shared representational space (Section 3.2) and further processed by a spatially-guided transformer, which facilitates the learning of layout-invariant representations (Section 3.3).

## 3.1 A GLOBAL DICTIONARY FOR HETEROGENEOUS ELECTRODE LAYOUTS

As illustrated in Figure 2A, a single EEG dataset often comprises a variety of electrode layouts, which may arise from the use of different EEG devices or from electrodes becoming non-functional or broken during data collection. To facilitate data preprocessing, we partition each dataset into subsets, each corresponding to one electrode layout. The channel coordinates $\mathcal{C}$ of each subset is defined as: $\mathcal{C} = \{N_{(i,e)}, X_{(i,e)}, \Pi_{(i,e)}\}, i = 1, \ldots, n, e = 1, \ldots, C_i$, where $N_{(i,e)}$ denotes the channel name, $X_{(i,e)}$ represents the channel type (e.g., EEG, EOG), and $\Pi_{(i,e)} \in \mathbb{R}^3$ is the 3D coordinate of the $e$-th channel in the $i$-th subset. Here, $C_i$ is the total number of channels in subset $i$, and $n$ is the total number of subsets. Details of the channel configurations across all datasets are provided in Appendix K.

**Global Channel Dictionary.** To ensure compatibility with heterogeneous layouts, we construct a global channel dictionary, as illustrated in Figure 2B, which incorporates 1,132 unique electrodes drawn from a wide range of EEG systems. These include the 10–20 system and its 10–10/10–5 extensions Jurcak et al. (2007), high-density arrays (e.g., EGI System Liu et al. (2017), Biosemi SystemKam et al. (2019)), numerical indices in MNE package Gramfort et al. (2014) (e.g., `EEG001-EEG074`), custom alphabetic in MNE package Gramfort et al. (2014) (e.g., `A1, B2`), region-based indices in MNE package (e.g., `E15-E256`), and standard reference points (e.g., `Ref, T3, M1`) Cobb et al. (1958). This global dictionary is represented as $\mathcal{C}_{\text{global}} = \{N_e, X_e, \Pi_e\}_{e=1}^{|c|}$, where $|c|$ is the total number of channels included in the dictionary; $N_e$ denotes the electrode name, $X_e$ represents the channel type, and $\Pi_e \in \mathbb{R}^3$ specifies the 3D coordinate of the $e$-th electrode.

**Mapping Electrodes into Global Dictionary.** As shown in Figure 2C, for the $i$-th subset, we define the set of available electrode indices as $\mathcal{I}_i = \{e \mid N_{(i,e)} \in \mathcal{C}_{\text{global}}, e = 1, \ldots, C_i\}$, where $\mathcal{I}_i$ denotes the index set of all available electrodes in the subset $i$. An electrode is considered available if it matches an entry in the global dictionary $\mathcal{C}_{\text{global}}$. Let the raw EEG data be $X = \{x_i \in \mathbb{R}^{C_i \times T_i}, i = 1, ..., n\}$, where $x_i$ is the EEG recording from the $i$-th subset with $T_i$ time points. With the obtained $\mathcal{I}_i$, we extract matched channels to form the input EEG data: $\bar{X} = \{\bar{x}_i \in \mathbb{R}^{|\mathcal{I}_i| \times T_i} \mid \bar{x}_i = x_i[\mathcal{I}_i, :], i = 1, ..., n\}$. Correspondingly, the 3D coordinates of available electrodes in the $i$-th subset are denoted as: $\mathbf{P}_i = [\Pi_e]_{e \in \mathcal{I}_i} \in \mathbb{R}^{|\mathcal{I}_i| \times 3}$.

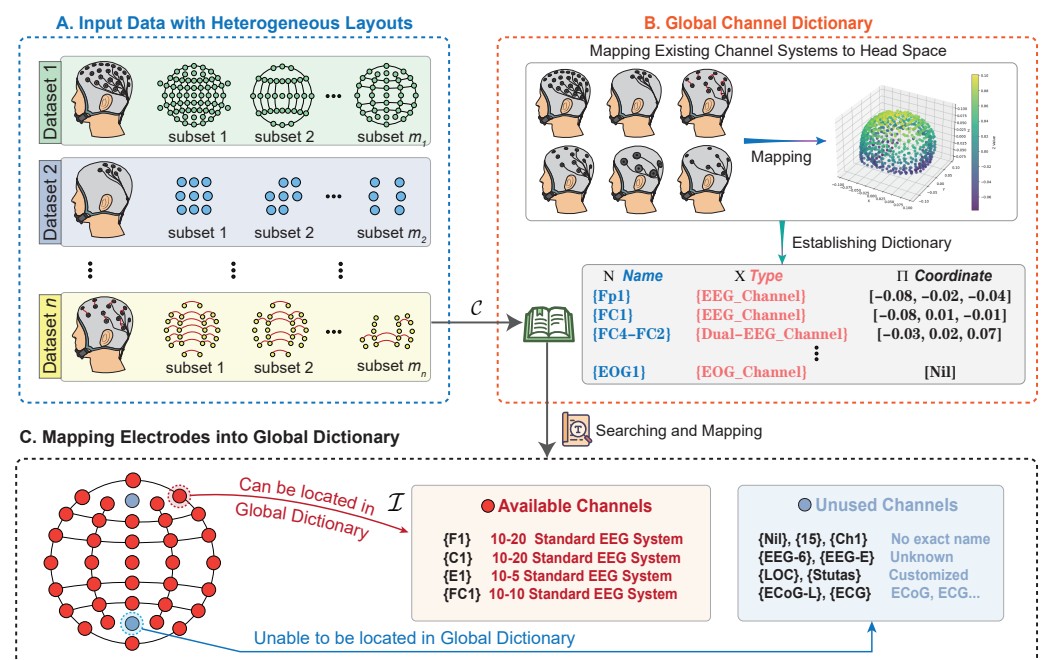

Figure 2: Mapping heterogeneous electrode layouts into a global channel dictionary. (A) Each EEG dataset may contain multiple electrode layouts. (B) A global dictionary is constructed to accommodate these heterogeneous layouts, recording each electrode's channel name, channel type, and 3D coordinates. (C) For a given electrode layout from a specific dataset, each electrode is labeled using the global dictionary. Electrodes that are not found in the global dictionary, such as those lacking identifiable channel names or originating from other modalities (e.g., ECG) are excluded.

## 3.2 COORDINATE-BASED SPATIAL-TEMPORAL EMBEDDING

As shown in Figure 1A, after the channel mapping stage, the EEG recordings with available channels $\bar{X}$ are fed into a temporal encoder, segmented into non-overlapping temporal windows of length $w$. For a subset $i$ with $C_i$ electrodes and $T_i$ time points, this results in $C_i \times \lfloor \frac{T_i}{w} \rfloor$ patches and can be denoted as: $\widetilde{X}_i = \left\{ \bar{x}_{(i,e),t} \in \mathbb{R}^w, e = 1, \ldots, C_i, \ t = 1, \ldots, \lfloor \frac{T_i}{w} \rfloor \right\}$.

As shown in Figure 1B, with the obtained coordinates $\mathbf{P}_i$ for subset $i$ , we apply a spatial MLP to project each electrode's 3D coordinate into the model embedding space:

$$\mathbf{S}_i = \mathrm{MLP}_{\mathrm{spatial}}(\mathbf{P}_i) \in \mathbb{R}^{C_i \times D}, \tag{1}$$

where $D$ is the hidden dimension, and $\mathbf{S}_{(i,e)} \in \mathbb{R}^D$ denotes the spatial embedding of electrode $e$. This spatial embedding is broadcast across the temporal dimension and concatenated with a learnable [CLS] token, and EEG patch tokens $\widetilde{X}_i$ are then added with the expanded spatial embeddings:

$$\mathbf{S}_i^{\mathrm{expanded}} = \left\{ \mathbf{s}_{(i,e)} \right\}_{e \in \mathcal{I}_i}^{\times T_i/w} \in \mathbb{R}^{C_i T_i / w \times D}, \tag{2}$$

$$\mathbf{X}_i^{\mathrm{spatial}} = \left[ \mathbf{x}_{\mathrm{CLS}}; \widetilde{X}_i \right] + \left[ \mathbf{0}; \mathbf{S}_i^{\mathrm{expanded}} \right] \in \mathbb{R}^{(1 + C_i T_i / w) \times D}. \tag{3}$$

To encode temporal information, we follow LaBraM by incorporating a learnable temporal embedding $\mathbf{T}_i$ and add it to the token sequence according to:

$$\mathbf{T}_i \in \mathbb{R}^{C_i T_i / w \times D}, \quad \mathbf{X}_i^{\mathrm{emb}} = \mathbf{X}_i^{\mathrm{spatial}} + [\mathbf{0}; \mathbf{T}_i]. \tag{4}$$

### 3.3 TEMPORAL-SLICE CHANNEL ATTENTION AND SPATIALLY-GUIDED TRANSFORMER

To explicitly model the spatial dependencies among electrodes, we further introduce two modules: (1) a temporal-slice channel attention module that captures electrode interactions within each temporal slice (Figure 1C), and (2) a spatially-guided Transformer that incorporates pairwise distances between electrodes into the attention weights, thereby enhancing the model's spatial awareness (Figure 1D).

**Temporal-Slice Channel Attention.** Given the embedded EEG sequence $\mathbf{X}_i^{\text{emb}} \in \mathbb{R}^{B \times C_i \times T_i/w \times D}$, we first reshape the patch tensor and subsequently apply an attention mechanism across the channels to capture spatial dependencies among different channels within each temporal slice. Formally,

$$\mathbf{X}_i^{(t)} \in \mathbb{R}^{B \cdot T_i/w \times C_i \times D}, \quad \widehat{\mathbf{X}}_i^{(t)} = \text{MultiHeadAttention}(\mathbf{X}_i^{(t)}, \mathbf{X}_i^{(t)}, \mathbf{X}_i^{(t)}). \tag{5}$$

The output is reshaped to a flattened patch sequence and concatenated with a zero [CLS] token:

$$\mathbf{X}_i^{\text{attn}} = \left[\mathbf{x}_{\text{CLS}}; \mathbf{X}_i^{\text{emb}}\right] \in \mathbb{R}^{B \times (1 + C_i T_i/w) \times D}. \tag{6}$$

where the [CLS] token is added at the beginning to locate the starting position of the embedding.

**Spatially-Guided Transformer.** In addition to the 3D coordinates of each electrode, pairwise distances provide complementary spatial information. As illustrated in Figure 1D, we integrate this spatial prior into the Transformer model to improve its spatial awareness. Specifically, given the 3D coordinate matrix $\mathbf{P}_i \in \mathbb{R}^{C_i \times 3}$ for subset $i$, we compute the pairwise differences between electrodes and transform these into a spatial bias embedding $\mathbf{B}$ using a shared MLP:

$$\Delta \mathbf{P}_i = \mathbf{p}_i(e_1) - \mathbf{p}_i(e_2), \quad \mathbf{B}^{(h)} = \text{MLP}_{\text{bias}}(\Delta \mathbf{P}_i) \in \mathbb{R}^{H \times C_i \times C_i}, \tag{7}$$

where $H$ is the number of attention heads and $\mathbf{B}^{(h)}$ encodes relative spatial relationships.

To align with the tokenized patch sequence, we expand the spatial bias embedding along the temporal axis to get $\mathbf{B}_{\text{ext}}^{(h)} \in \mathbb{R}^{H \times C_i T_i/w \times C_i T_i/w}$. The expanded spatial bias is then inserted into the Transformer attention map as a relative bias term:

$$\mathbf{B}_{\text{rel}}[0, :, 1:, 1:] = \mathbf{B}_{\text{ext}}^{(h)}, \quad \mathbf{B}_{\text{rel}}[0, :, 0, :] = \mathbf{B}_{\text{rel}}[0, :, :, 0] = 0. \tag{8}$$

The bias is applied only to patch tokens, while the [CLS] token is excluded by setting it to zero.

## 4 HEAR DATASET

In this section, we present the HEAR dataset, which has been compiled from 20 publicly available EEG datasets encompassing a diverse array of electrode configurations. Table 2 provides an overview of the included datasets, detailing their names, durations, electrode systems, and channel configurations. The HEAR dataset spans a broad range of experimental protocols and electrode systems, comprising a total of 8,782 hours of EEG recordings across more than 150 unique channel layouts. Detailed descriptions of the original datasets are provided in Appendix J, and comprehensive statistical analyses of the postprocessed dataset are presented in Appendix 12, 13, and 14.

## 5 EXPERIMENTAL RESULTS

In this section, we first present the experimental setting. We then introduce the experimental results of the HEAR model on five downstream tasks to demonstrate its generalizability and scalability. Subsequently, we present an ablation study to validate the effectiveness of different components within HEAR. Finally, we conduct a series of analyses to demonstrate the enhanced spatial awareness and robustness of the HEAR model.

Table 2: Overview of the HEAR dataset, compiled from 20 publicly available EEG datasets with a wide range of electrode configurations. "#Configs" denotes the number of distinct channel configurations in each dataset; "#Channels" refers to the number of channels per configuration.

| | Dataset Name | Hours | Electrode System | #Configs | #Channels |
|---|---|---|---|---|---|
| **Pretraining** | *Migrainedb* | 21.2 | 10–5 system + multi-modal | 1 | 128 |
| | *PhysioNetP300* | 2.3 | 10–20 system | 1 | 32 |
| | *OpenBMI* | 91.6 | 10–10 system + multi-modal | 1 | 62 |
| | *EEGMAT* | 2.4 | 10–20 system + multi-modal | 1 | 20–21 |
| | *KaggleERN* | 30.0 | 10–10 system | 1 | 62 |
| | *TUEP* | 631.8 | 10–20 system | 34 | 21–40 |
| | *TUEV* | 148.7 | 10–20 system | 14 | 21–32 |
| | *HMCSleep* | 582.5 | bipolar system (sleep) | 1 | 6–8 |
| | *SleepEDFx* | 3,849.0 | bipolar system | 2 | 2–7 |
| | *TUAB* | 1,137.3 | 10–20 system | 17 | 21–40 |
| | *CHB-MIT* | 1,060.9 | 10–20 bipolar system | 12 | 22–38 |
| | *TUSL* | 27.6 | 10–20 system | 12 | 27–41 |
| | *CAP-Sleep* | 1,004.5 | 10–20 bipolar system | 50 | 5-36 |
| **Testing** | *BCI-IV-1* | 3.7 | 10–10 system | 1 | 59 |
| | *BCI-IV-2B* | 26.3 | 10–20 system | 1 | 3 |
| | *EEGMMIDB* | 48.5 | 10–10 system | 1 | 64 |
| | *LargeMI* | 59.4 | 10–20 system | 1 | 23 |
| | *SHUDB* | 12.4 | 10–10 system | 1 | 32 |
| | *BCI-IV-2A* | 13.4 | 10–20 system | 1 | 22 |
| | *HGD* | 28.7 | high-density 10–5 system | 1 | 128 |

## 5.1 EXPERIMENTAL SETTINGS

**Model Configurations.** These models differ in architectural depth, ranging from 6 to 12 layers, with the number of attention heads increasing from 4 to 8, resulting in a total of 3.1M and 6.0M parameters, respectively. Both variants employ a fixed vocabulary size of 2048. Pretraining is performed on 8×NVIDIA A6000 GPUs, and learning rates are selected empirically to ensure stable convergence.

**Evaluation Protocol.** All experiments adhere to a standardized training and testing protocol to ensure consistent and reproducible evaluation. Datasets are partitioned into training, validation, and test subsets using a fixed ratio of 3:1:1. For each run, the model achieving the best validation result is selected for evaluation on the test set to report final results. Each experiment is repeated with three random seeds (0, 1, and 2), and we report the mean and standard deviation across these runs. To comprehensively evaluate model performance, we compute three metrics: Balanced Accuracy, Weighted F1, and Macro F1, in accordance with relevant literature Brodersen et al. (2010); Lipton et al. (2014); Grandini et al. (2020). All datasets are preprocessed using a consistent pipeline, with detailed procedures provided in Appendix I.

**Baseline Models.** We compare our models against four publicly available EEG foundation models: BENDR Kostas et al. (2021), BIOT Yang et al. (2024), LaBraM Jiang et al. (2024), and EEGPT Wang et al. (2024a). All baseline models are evaluated using the same protocol described above.

## 5.2 EXPERIMENTAL RESULTS ON DOWNSTREAM TASKS

As presented in Table 3, our models exhibit superior performance across a diverse set of EEG decoding tasks. Notably, HEAR-tiny—with only 3.1M parameters—consistently outperforms state-of-the-art EEG foundation models on five benchmark datasets, underscoring the significantly enhanced representational power provided by our proposed HEAR model and dataset. Although HEAR-base achieves the highest overall performance across all settings, the gains over HEAR-tiny are not substantial, likely due to saturation of the available training data. Therefore, we limit our model size to the base version in subsequent experiments, and will investigate larger models in future work as more training data becomes available.

Table 3: Comparison of different EEG foundation models. Evaluations on the *EEGMMIDB* and *BCI-IV-1* datasets are excluded for the EEGPT and LaBraM models, respectively, as these datasets were used during their pretraining. The **best** results are highlighted in bold, and second-best results are underlined.

| Datasets | Methods | Balanced Accuracy | Weighted F1 | Macro F1 |
|---|---|---|---|---|
| *BCI-IV-2B* | BIOT [3.2M] | 0.5524 ± 0.0101 | 0.5516 ± 0.0101 | 0.5516 ± 0.0101 |
| | BENDR [4.0M] | 0.6806 ± 0.0067 | 0.6801 ± 0.0067 | 0.6801 ± 0.0067 |
| | LaBraM [5.8M] | 0.6610 ± 0.0106 | 0.6608 ± 0.0105 | 0.6607 ± 0.0105 |
| | EEGPT [25.3M] | 0.6893 ± 0.0090 | 0.6890 ± 0.0089 | 0.6890 ± 0.0089 |
| | **HEAR-tiny** [3.1M] | 0.7187 ± 0.0092 | 0.7163 ± 0.0092 | 0.7164 ± 0.0092 |
| | **HEAR-base** [6.0M] | **0.7213 ± 0.0094** | **0.7192 ± 0.0092** | **0.7193 ± 0.0092** |
| *LargeMI* | BIOT [3.2M] | 0.4709 ± 0.0088 | 0.4944 ± 0.0066 | 0.4685 ± 0.0080 |
| | BENDR [4.0M] | 0.6960 ± 0.0068 | 0.7220 ± 0.0047 | 0.7070 ± 0.0055 |
| | LaBraM [5.8M] | 0.5125 ± 0.0108 | 0.5455 ± 0.0099 | 0.5155 ± 0.0115 |
| | EEGPT [25.3M] | 0.6396 ± 0.0129 | 0.6705 ± 0.0121 | 0.6474 ± 0.0136 |
| | **HEAR-tiny** [3.1M] | 0.7104 ± 0.0165 | 0.7145 ± 0.0141 | 0.7129 ± 0.0138 |
| | **HEAR-base** [6.0M] | **0.7381 ± 0.0194** | **0.7388 ± 0.0182** | **0.7401 ± 0.0180** |
| *SHUDB* | BIOT [3.2M] | 0.5884 ± 0.0072 | 0.5874 ± 0.0079 | 0.5873 ± 0.0079 |
| | BENDR [4.0M] | 0.6276 ± 0.0094 | 0.6264 ± 0.0097 | 0.6263 ± 0.0098 |
| | LaBraM [5.8M] | 0.6266 ± 0.0138 | 0.6265 ± 0.0139 | 0.6264 ± 0.0138 |
| | EEGPT [25.3M] | 0.6074 ± 0.0138 | 0.6070 ± 0.0138 | 0.6070 ± 0.0138 |
| | **HEAR-tiny** [3.1M] | 0.6307 ± 0.0130 | 0.6282 ± 0.0137 | 0.6283 ± 0.0137 |
| | **HEAR-base** [6.0M] | **0.6350 ± 0.0140** | **0.6290 ± 0.0198** | **0.6293 ± 0.0193** |
| *EEGMMIDB* | BIOT [3.2M] | 0.3301 ± 0.0096 | 0.3274 ± 0.0089 | 0.3272 ± 0.0091 |
| | BENDR [4.0M] | 0.5368 ± 0.0097 | 0.5364 ± 0.0125 | 0.5361 ± 0.0120 |
| | LaBraM [5.8M] | 0.5033 ± 0.0072 | 0.5032 ± 0.0065 | 0.5029 ± 0.0067 |
| | **HEAR-tiny** [3.1M] | 0.5625 ± 0.0121 | 0.5668 ± 0.0125 | 0.5667 ± 0.0125 |
| | **HEAR-base** [6.0M] | **0.5651 ± 0.0134** | **0.5688 ± 0.0139** | **0.5688 ± 0.0139** |
| *BCI-IV-1* | BIOT [3.2M] | 0.5667 ± 0.0229 | 0.5730 ± 0.0216 | 0.5631 ± 0.0212 |
| | BENDR [4.0M] | 0.4980 ± 0.0278 | 0.4935 ± 0.0310 | 0.4577 ± 0.0317 |
| | EEGPT [25.3M] | 0.5410 ± 0.0389 | 0.5504 ± 0.0400 | 0.5214 ± 0.0532 |
| | **HEAR-tiny** [3.1M] | 0.5758 ± 0.0437 | **0.5738 ± 0.0400** | 0.5645 ± 0.0448 |
| | **HEAR-base** [6.0M] | **0.5799 ± 0.0370** | 0.5737 ± 0.0437 | **0.5648 ± 0.0441** |

We further investigate the impact of pretraining data size on model performance. As shown in Figure 3, increasing the amount of data used during pretraining leads to steady improvements across evaluation metrics, indicating that large-scale pretraining effectively enhances the generalization capability of our model. These results underscore the favorable scalability of HEAR with larger datasets.

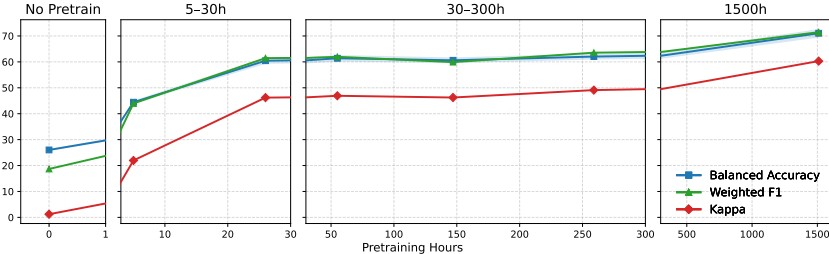

Figure 3: Experimental results on scaling the pretraining data size. The model is evaluated on the LargeMI dataset using Balanced Accuracy, Weighted F1, and Kappa metrics.

## 5.3 ABLATION STUDIES

To investigate the contributions of individual components in the proposed HEAR model, we conduct an ablation study based on four model variants: **A:** without the *Spatial Embedding*, **B:** without the *Temporal-Slice Channel Attention*, **C:** without the *Temporal-Slice Channel Attention* and the *Spatially-Guided Transformer*, and **D:** the full model.

As shown in Table 4, the ablation study reveals the complementary contributions of each architectural module. Removing the Spatial Embedding module leads to a moderate decline in performance, indicating its role in capturing spatial priors. More significant degradation is observed when both the Temporal-Slice Channel Attention and the Spatially-Guided Transformer are ablated, highlighting the importance of spatiotemporal interaction modeling in EEG signal representation learning. The full model consistently achieves the best performance across all datasets, validating the necessity of jointly modeling spatial structure and temporal dynamics. These results demonstrate that each component contributes non-trivially to the overall effectiveness of the architecture, and their integration enables robust generalization across EEG datasets.

Table 4: Results of the ablation study. The **best** results are highlighted in bold.

| Datasets | A: w/o | B: w/o | C: w/o | D: with all |
|---|---|---|---|---|
| *BCI-IV-1* | 0.5674±0.0334 | 0.5494±0.0325 | 0.5654±0.0343 | **0.5758±0.0437** |
| *BCI-IV-2B* | 0.7150±0.0093 | 0.6789±0.0090 | 0.6804±0.0087 | **0.7187±0.0092** |
| *LargeMI* | 0.6882±0.0203 | 0.4573±0.0177 | 0.4291±0.0180 | **0.7104±0.0165** |
| *SHUDB* | 0.5716±0.0392 | 0.5131±0.0105 | 0.5111±0.0094 | **0.6307±0.0130** |
| *EEGMMIDB* | 0.5299±0.0249 | 0.3193±0.0128 | 0.3188±0.0159 | **0.5625±0.0121** |

## 5.4 VISUALIZATION OF CHANNEL ACTIVATIONS

To further investigate how HEAR models spatial information, we visualize the attention weights from its spatial attention layer across the training process. Specifically, we extract the normalized attention scores for each EEG channel over 50 training epochs and present them as heatmaps in Figure 4 for two downstream datasets: BCI-IV-1 and EEGMMIDB. For each dataset, we also provide a topographic projection of channel-wise activation from the final epoch, offering an interpretable spatial perspective on the learned attention distribution.

HEAR exhibits stable and structured channel preferences over time. On BCI-IV-1, HEAR progressively concentrates its attention on motor-related regions, including FC3, FC4, and C3/C4, which align well with known sensorimotor cortices. This adaptation suggests that HEAR effectively prioritizes task-relevant spatial features Brake et al. (2024). In contrast, for the EEGMMIDB dataset, which involves cognitive load estimation Forenzo et al. (2023), the model's attention is more broadly distributed, with heightened activation observed over frontal and prefrontal electrodes (e.g., Fp1, AF3, F3, Fz) Schalk et al. (2004). More visualization results are presented in Appendix Figure 8.

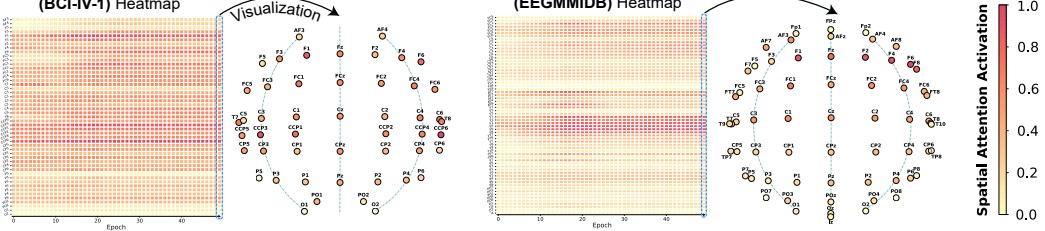

Figure 4: Channel activation visualization.

## 5.5 CHANNEL ROBUSTNESS ON UNSEEN ELECTRODES

To evaluate the robustness of the proposed model under layout change, we tested the model's performance on unseen electrodes on the HGD dataset. HGD is a completely new dataset that was

not used during the pretraining stage, and only appears during fine-tuning to support the cross-layout experiment. As shown in Figure 5(a), the HGD dataset contains 128 electrodes. In the pretraining dataset, the electrode layout only covers the electrode positions indicated by the blue dots in the figure, while evaluation is performed on a disjoint set of unseen electrodes (red triangles). We thereby simulate the situation under realistic variability in electrode configurations. The detailed channel layouts are presented in Appendix Table 11.

Figure 5(b) shows the accuracy across 11 distinct, evenly sampled channel subsets in the evaluated unseen electrodes. Across all settings, HEAR maintains a noticeable and stable advantage over the baseline EEG FMs. The results collectively highlight the model's robustness to unseen electrode configurations. The performance gap widens as the number of available channels increases, suggesting that HEAR effectively leverages richer spatial information under configuration variation.

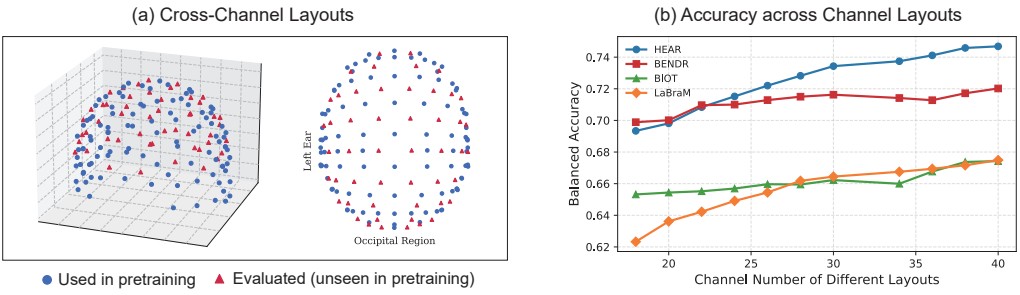

Figure 5: Experimental results of channel robustness on unseen electrodes using the HGD dataset.

## 5.6 ZERO-SHOT GENERALIZATION TO UNSEEN LAYOUTS

To further evaluate HEAR's zero-shot generalization ability on unseen layout, we partition the electrodes of BCI-IV-2B dataset into two non-overlapping subsets, forming Layout 1 (L1) and Layout 2 (L2). During fine-tuning stage, the model is trained on only one layout, and evaluated on the other at inference stage.

We report both transfer directions (L1→L2 and L2→L1) to assess robustness under unseen layouts. As shown in Table 5, across both transfer directions, Ours-tiny achieves the best performance, indicating strong zero-shot generalization to unseen layouts. EEGPT, which has the largest parameter count among the baselines, attains the second-best results in both directions. Overall, these results support that HEAR's heterogeneous representation improves robustness under layout shifts at inference time.

Table 5: Experimental results of zero-shot layout transfer on BCI-IV-2B dataset. The **best** results are highlighted in bold, and second-best results are underlined.

| Methods | L1→L2 (ACC) | L1→L2 (F1) | L2→L1 (ACC) | L2→L1 (F1) |
|---|---|---|---|---|
| BENDR | 0.5757 ± 0.0128 | 0.5752 ± 0.0128 | 0.5840 ± 0.0172 | 0.5837 ± 0.0168 |
| BIOT | 0.5130 ± 0.0106 | 0.4985 ± 0.0230 | 0.5170 ± 0.0132 | 0.5010 ± 0.0182 |
| LaBraM | 0.4942 ± 0.0189 | 0.4733 ± 0.0142 | 0.4971 ± 0.0136 | 0.4971 ± 0.0136 |
| EEGPT | 0.6292 ± 0.0146 | 0.6283 ± 0.0144 | 0.6292 ± 0.0146 | 0.6283 ± 0.0144 |
| **Ours-tiny** | **0.7190 ± 0.0065** | **0.7139 ± 0.0048** | **0.7071 ± 0.0101** | **0.7014 ± 0.0084** |

## 6 CONCLUSION

In this paper, we present HEAR, an EEG foundation model specifically designed to address the heterogeneity inherent in electrode layouts. By assembling a large-scale EEG dataset encompassing diverse and heterogeneous setups, and introducing a series of architectural innovations, HEAR achieves robust neural decoding performance across arbitrary and previously unseen EEG layouts. Extensive experimental results highlight the superior generalization, adaptability, and robustness of the proposed model. Consequently, HEAR paves the way for scalable and unified EEG modeling, accelerating progress in both fundamental neuroscience research and translational BCI applications.

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

# APPENDIX

## A    ETHICS STATEMENT

This work adheres to the ICLR Code of Ethics. In this study, no human subjects or animal experimentation were involved. All datasets used, including Migrainedb, PhysioNetP300, OpenBMI, EEGMAT, KaggleERN, TUEP, TUEV, HMCSleep, SleepEDFx, TUAB, CHB-MIT, TUSL, CAP-Sleep, BCI-IV-1, BCI-IV-2B, EEGMMIDB, LargeMI, SHUDB, BCI-IV-2A, and HGD, were sourced in compliance with relevant usage guidelines, ensuring no violation of privacy. We have taken care to avoid any biases or discriminatory outcomes in our research process. No personally identifiable information was used, and no experiments were conducted that could raise privacy or security concerns. We are committed to maintaining transparency and integrity throughout the research process.

## B    REPRODUCIBILITY STATEMENT

We have made every effort to ensure that the results presented in this paper are reproducible. All code and datasets have been made publicly available in an anonymous repository to facilitate replication and verification. The experimental setup, including training steps, model configurations, and hardware details, is described in detail in the paper. We have also provided a full description of HEAR to assist others in reproducing our experiments.

Additionally, Migrainedb, PhysioNetP300, OpenBMI, EEGMAT, KaggleERN, TUEP, TUEV, HM-CSleep, SleepEDFx, TUAB, CHB-MIT, TUSL, CAP-Sleep, BCI-IV-1, BCI-IV-2B, EEGMMIDB, LargeMI, SHUDB, BCI-IV-2A, HGD are publicly available, ensuring consistent and reproducible evaluation results.

We believe these measures will enable other researchers to reproduce our work and further advance the field.

## C    LLM USAGE

Large Language Models (LLMs) were used to aid in the writing and polishing of the manuscript. Specifically, we used an LLM to assist in refining the language, improving readability, and ensuring clarity in various sections of the paper. The model helped with tasks such as sentence rephrasing, grammar checking, and enhancing the overall flow of the text.

It is important to note that the LLM was not involved in the ideation, research methodology, or experimental design. All research concepts, ideas, and analyses were developed and conducted by the authors. The contributions of the LLM were solely focused on improving the linguistic quality of the paper, with no involvement in the scientific content or data analysis.

The authors take full responsibility for the content of the manuscript, including any text generated or polished by the LLM. We have ensured that the LLM-generated text adheres to ethical guidelines and does not contribute to plagiarism or scientific misconduct.

## D  BACKGROUND AND RELATED WORK

Electroencephalography (EEG) offers a dynamic and non-invasive means of monitoring brain activity by capturing electrical signals from the cerebral cortex. Its portability and real-time capabilities make it an effective tool in neuroscience research and practical applications, particularly in brain-computer interfaces (BCIs). Despite its widespread adoption, the performance of EEG-based algorithms is fundamentally constrained by inherent challenges, including a low signal-to-noise ratio (SNR), substantial inter-subject variability, and most notably, task-dependent fluctuations in recorded signals Craik et al. (2019). These fluctuations play a crucial role in EEG analysis as they capture task-related neural dynamics, but they also introduce substantial variability across recordings, making it challenging to develop models with robust generalization. While numerous machine learning approaches have demonstrated success on individual datasets Lawhern et al. (2018), their performance tends to degrade substantially in real-world scenarios due to domain shifts, distribution mismatches, and individual differences in EEG data. This lack of generalizability restricts the applicability of models trained on specific datasets, presenting a critical obstacle for the deployment of robust EEG-based systems.

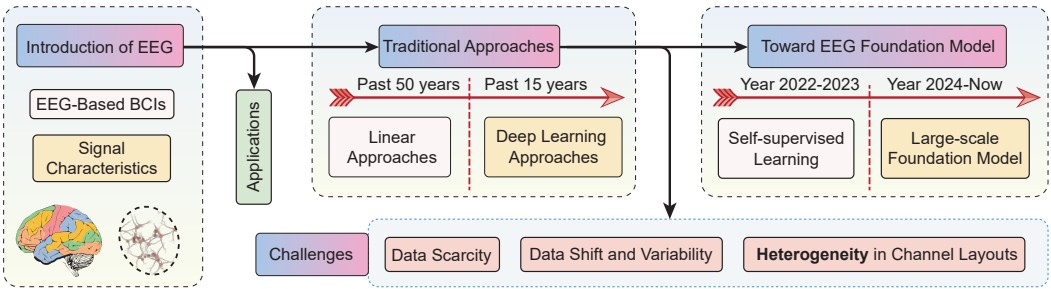

Figure 6: Evolution of EEG decoding algorithms over the past two decades, highlighting the transition from traditional approaches to deep learning, self-supervised learning, and the development of the EEG foundation model.

**Self-supervised learning of EEG signals:** In recent years, self-supervised learning (SSL) has emerged as a powerful technique in EEG analysis, showing promise for mitigating the poor generalization of models trained on single EEG datasets Rafiei et al. (2022). Techniques such as masked autoencoders (MAE) have shown considerable success in EEG analysis Kostas et al. (2021), learning effective representations by reconstructing masked signal patches Rafiei et al. (2022).

**Large-scale EEG foundation models:** Building upon these advances, EEG foundation models (EEG FMs) Lai et al. (2025) leverage large-scale EEG data to learn unified representations that generalize across a wide range of tasks and subjects Jiang et al. (2024); Wang et al. (2024a). By pretraining on extensive unlabeled EEG data and fine-tuning for specific tasks, EEG FMs reduce the dependency on labeled data and significantly enhance generalization, consistently outperforming deep learning models trained on individual datasets. This ability to capture transferable knowledge across diverse EEG data makes EEG FMs a promising approach for EEG decoding. In response to the increasing demand for scalable and generalizable EEG analysis, a number of EEG foundation models (FMs) have recently been proposed, leveraging large-scale pretraining to enhance downstream performance and cross-domain adaptability. In general, there are nine publicly available EEG FMs: *BIOT* Yang et al. (2024), *BENDR* Kostas et al. (2021), *Brant* Zhang et al. (2023), *BrainBERT* Wang et al. (2023), *LaBraM* Jiang et al. (2024), *Neuro-GPT* Cui et al. (2024), *EEGPT* Wang et al. (2024a), *FoME* Shi et al. (2024), and *CBraMod* Wang et al. (2024b).

These models explore diverse architectural designs and pretraining strategies to improve the generalizability and scalability of EEG-based neural decoding. *BIOT* leverages a novel biosignal tokenization scheme to transform EEG recordings of varied lengths, channel configurations, and missing values

into unified "biosignal sentences," enabling robust cross-dataset learning via a linear transformer architecture Yang et al. (2024). *BENDR* introduces a masked predictive learning framework inspired by wav2vec, using convolutional encoders and transformer decoders to model long-range temporal dependencies in raw EEG data, which is shown to transfer effectively across tasks Kostas et al. (2021). *Brant*, tailored for intracranial signals, jointly models long-term temporal patterns and spatial channel correlations via dual temporal–spatial transformer encoders, and incorporates frequency-aware embeddings to improve forecasting and seizure detection capabilities Zhang et al. (2023). *BrainBERT* formulates EEG spectrogram reconstruction as a self-supervised learning task, employing masked modeling and adaptive content-aware losses to learn transferable contextual representations of neural activity Wang et al. (2023). *LaBraM* scales pretraining across 2,500+ hours of EEG using a neural tokenizer and transformer backbone to encode patch-wise EEG fragments, facilitating masked token prediction across datasets with diverse spatial and temporal layouts Jiang et al. (2024). *Neuro-GPT* introduces a GPT-style decoder architecture, pairing an EEG encoder with autoregressive masked prediction, to learn causally structured temporal embeddings across large-scale EEG corpora Cui et al. (2024). *EEGPT* designs a dual self-supervised strategy combining spatio-temporal representation alignment and reconstruction objectives, supported by a hierarchical transformer to separately encode spatial and temporal information for enhanced generalization in multi-paradigm EEG decoding Wang et al. (2024a). Lastly, *FoME* presents a large-scale pretraining pipeline featuring time-frequency fusion and adaptive temporal-lateral attention scaling (ATLAS), achieving robust performance across classification and forecasting tasks on heterogeneous EEG and iEEG datasets Shi et al. (2024).

# E FORMULA SUPPLEMENT

In this section, we present supplement formulas of the methodology.

## E.1 HETEROGENEOUS SELF-SUPERVISED PRETRAINING

**Neural Representation:** Following the pretraining architecture from LaBraM Jiang et al. (2024), all the patch representations are quantized into the neural embeddings via a neural codebook $\mathcal{V} = \{v_i \mid i = 1, \ldots, K\} \in \mathcal{R}^{K \times D}$:

$$z_{(i,j)} = \arg\min \left\| \ell_2 \left( p_{(i,j)} \right) - \ell_2 \left( v_j \right) \right\|_2 \tag{9}$$

where $\ell_2(x)$ denotes the $\ell_2$-normalization of the vector $x$, $z_{(i,k)}$ denotes the obtained quantized vector.

The quantization loss can thus be computed by:

$$\mathcal{L}_{\mathcal{Q}} = \sum_{i=1}^{n} \sum_{j=1}^{C_i \frac{T_i}{w}} \left\| sg \left[ \ell_2 \left( \widehat{x}_{(i,j)} \right) \right] - \ell_2 \left( v_{z_{(i,j)}} \right) \right\|^2 + \left\| \ell_2 \left( \widehat{x}_{(i,j)} \right) - sg \left[ \ell_2 \left( v_{z_{(i,j)}} \right) \right] \right\| \tag{10}$$

where $\widehat{x}_{(i,j)}$ denotes the patch representations of EEG data, $v_{z_{(i,j)}}$ is the codebook vector corresponding to the quantized token, and $sg$ refers to the stop-gradient operation.

**Fourier Spectrum Prediction:** The reconstruction loss focuses on reconstructing the Fourier spectrum (amplitude and phase) from quantized patches and can be computed by:

$$\mathcal{L}_{\mathcal{S}} = \sum_{i=1}^{n} \sum_{j=1}^{C_i^i \frac{T_i}{v}} \left\| \widetilde{A}_{(i,j)} - A_{(i,j)} \right\|^2 + \left\| \widetilde{\psi}_{(i,j)} - \psi_{(i,j)} \right\| \tag{11}$$

where $\widetilde{A}_{(i,j)}$ and $A_{(i,j)}$ are the predicted and ground truth amplitude values, and $\widetilde{\psi}_{(i,j)}$ and $\psi_{(i,j)}$ are the predicted and ground truth phase values of the Fourier spectrum.

# F ADDITIONAL RESULTS

## F.1 ADDITIONAL DOWNSTREAM RESULTS ON TUAB DATASET

We evaluate the HEAR performance on the TUAB dataset. All methods follow the exact fine-tuning protocol used in the main paper. As shown in Table 6, HEAR achieves the best performance on all metrics. Notably, HEAR-tiny surpasses EEGPT while using fewer parameters.

Table 6: Results on the TUAB dataset.

| Methods | Balanced Accuracy | Weighted-F1 | Macro-F1 |
|---|---|---|---|
| BENDR | 0.9116 ± 0.0067 | 0.9568 ± 0.0018 | 0.9266 ± 0.0031 |
| BIOT | 0.7878 ± 0.0035 | 0.8979 ± 0.0022 | 0.8200 ± 0.0025 |
| LaBraM | 0.7987 ± 0.0040 | 0.8951 ± 0.0004 | 0.8189 ± 0.0023 |
| EEGPT | 0.9314 ± 0.0019 | 0.9642 ± 0.0005 | 0.9397 ± 0.0005 |
| **HEAR-tiny** | **0.9441** ± 0.0303 | **0.9692** ± 0.0134 | **0.9482** ± 0.0231 |
| **HEAR-base** | **0.9620** ± 0.0148 | **0.9807** ± 0.0082 | **0.9677** ± 0.0134 |

## F.2 ADDITIONAL TRANSFER LEARNING EXPERIMENT

We study two transfer scenarios: (i) cross-subject on BCI-IV-2B, and (ii) cross-task on LargeMI (hand MI ↔ feet MI). In each case, training and validation come from the same source domain (2:1 split), while testing uses a disjoint target domain (unseen subjects/tasks). The fine-tuning protocol is identical to the main paper, and we compare representative EEG-FMs and HEAR.

As shown in Table 7 and Table 8, HEAR consistently outperforms baselines across transfer types.

Table 7: Cross-subject transfer on BCI-IV-2B.

| Pair | BENDR | BIOT | LaBraM | EEGPT | **HEAR-tiny** | **HEAR-base** |
|---|---|---|---|---|---|---|
| S1→S2 | 0.5344 | 0.4979 | 0.5104 | 0.5562 | 0.7828 | 0.7732 |
| S1→S3 | 0.5817 | 0.5260 | 0.5106 | 0.5788 | 0.5000 | 0.6747 |
| S2→S1 | 0.5543 | 0.5033 | 0.5130 | 0.5598 | 0.7614 | 0.7522 |
| S2→S3 | 0.5529 | 0.5163 | 0.4933 | 0.5231 | 0.6477 | 0.5000 |
| S3→S1 | 0.6022 | 0.5293 | 0.5293 | 0.6293 | 0.5538 | 0.5423 |
| S3→S2 | 0.5458 | 0.4938 | 0.5417 | 0.5531 | 0.5500 | 0.5692 |
| Average | 0.5619 | 0.5111 | 0.5164 | 0.5667 | **0.6326** | **0.6353** |

Table 8: Cross-task transfer on LargeMI.

| Methods | T1→T2 (ACC) | T1→T2 (F1) | T2→T1 (ACC) | T2→T1 (F1) |
|---|---|---|---|---|
| BENDR | 0.6519 ± 0.0152 | 0.6470 ± 0.0180 | 0.6536 ± 0.0071 | 0.6536 ± 0.0071 |
| BIOT | 0.5208 ± 0.0055 | 0.5180 ± 0.0069 | 0.5158 ± 0.0098 | 0.5046 ± 0.0099 |
| LaBraM | 0.6165 ± 0.0129 | 0.6152 ± 0.0138 | 0.5556 ± 0.0095 | 0.5462 ± 0.0101 |
| EEGPT | 0.6712 ± 0.0095 | 0.6664 ± 0.0124 | 0.6625 ± 0.0018 | 0.6606 ± 0.0011 |
| **HEAR-tiny** | **0.7244** ± 0.0521 | **0.7230** ± 0.0523 | 0.6069 ± 0.1467 | 0.5082 ± 0.2366 |
| **HEAR-base** | 0.7027 ± 0.1141 | 0.6692 ± 0.1864 | **0.6970** ± 0.1131 | **0.6623** ± 0.1842 |

## F.3 COMPARISON WITH THE SPECIALIZED SMALL MODEL

We compare pretrained HEAR against EEGNet trained from scratch on two representative MI datasets (BCI-IV-1, BCI-IV-2B). We report accuracy together with model size (parameters) and compute (FLOPs).

As shown in Table 9, pretrained HEAR provides significant accuracy gains over EEGNet on both datasets, reflecting the benefits of heterogeneous pretraining and spatial priors. This comes with higher FLOPs, suggesting future directions like knowledge distillation or sparse inference.

Table 9: Comparison results between the specialized small model and HEAR.

| Methods | BCI-IV-1 (ACC) | BCI-IV-2B (ACC) | Params | FLOPs |
|---------|----------------|-----------------|--------|-------|
| EEGNet | 0.5458 ± 0.0302 | 0.6655 ± 0.0430 | 2.26k | 4.39M |
| **HEAR-tiny** | **0.5758** ± 0.0437 | **0.7187** ± 0.0092 | 3.1M | 1.23G |
| **HEAR-base** | **0.5799** ± 0.0370 | **0.7213** ± 0.0094 | 6.0M | 2.45G |

# G ADDITIONAL EXPERIMENT SETTINGS

## G.1 CHANNEL CONFIGURATIONS OF COMPARISON FOUNDATION MODELS

Table 10 summarizes the electrode configurations adopted by each compared EEG foundation model. The diversity in channel selection reveals notable differences in the spatial priors and pretraining assumptions across models.

Table 10: Electrode configurations used by the comparison EEG foundation models.

| Model | Pretrained Channel Names |
|-------|--------------------------|
| **BENDR** | FPZ, FP1, FP2, F3, F4, FZ, C3, C4, CZ, P3, P4, PZ, O1, O2, OZ, T7, T8, P7, P8 |
| **EEGPT** | FP1, FPZ, FP2, AF7, AF3, AF4, AF8, F7, F5, F3, F1, FZ, F2, F4, F6, F8, FT7, FC5, FC3, FC1, FCZ, FC2, FC4, FC6, FT8, T7, C5, C3, C1, CZ, C2, C4, C6, T8, TP7, CP5, CP3, CP1, CPZ, CP2, CP4, CP6, TP8, P7, P5, P3, P1, PZ, P2, P4, P6, P8, PO7, PO5, PO3, POZ, PO4, PO6, PO8, O1, OZ, O2 |
| **LaBraM** | FP1, FPZ, FP2, AF9, AF7, AF5, AF3, AF1, AFZ, AF2, AF4, AF6, AF8, AF10, F9, F7, F5, F3, F1, FZ, F2, F4, F6, F8, F10, FT9, FT7, FC5, FC3, FC1, FCZ, FC2, FC4, FC6, FT8, FT10, T9, T7, C5, C3, C1, CZ, C2, C4, C6, T8, T10, TP9, TP7, CP5, CP3, CP1, CPZ, CP2, CP4, CP6, TP8, TP10, P9, P7, P5, P3, P1, PZ, P2, P4, P6, P8, P10, PO9, PO7, PO5, PO3, PO1, POZ, PO2, PO4, PO6, PO8, PO10, O1, OZ, O2, O9, CB1, CB2, IZ, O10, T3, T5, T4, T6, M1, M2, A1, A2, CFC1–CFC8, CCP1–CCP8, T1, T2, FTT9h, TTP7h, TPP9h, FTT10h, TPP8h, TPP10h |
| **BIOT** | C3-A2, C4-A1 (Sleep); 16 EEG montages (PREST, private); ECG (non-EEG) |

BENDR adopts a compact configuration with only 19 channels, primarily concentrated along the midline and bilateral frontal, central, parietal, and occipital regions. This minimalist setup prioritizes compatibility with commonly available EEG systems but limits spatial granularity. In contrast, EEGPT leverages a richer 64-channel subset conforming to the 10-20 system, encompassing broad coverage over frontal, temporal, central, parietal, and occipital areas. This setup facilitates the extraction of more nuanced spatial features, contributing to its robust generalization.

LaBraM employs an extremely dense configuration, incorporating over 120 channels including extended 10-10 and 10-5 system labels, as well as custom montage identifiers (e.g., CFC, CCP, FTT, TPP). Such comprehensive coverage enables fine-grained spatial learning, but it also introduces significant variability and potential overfitting risks when transferring to downstream datasets with fewer channels.

For BIOT, the situation differs: its pretraining leverages clinical datasets such as SHHS and PREST, where EEG signals are recorded using limited montages (e.g., bipolar derivations like C3-A2 and C4-A1). This restricted spatial resolution poses inherent challenges for transferability to tasks requiring high-density representations. Additionally, a portion of BIOT's pretraining dataset includes ECG data, which does not directly align with EEG spatial priors.

In summary, the channel configurations reflect trade-offs between generalization and granularity. HEAR is designed to flexibly accommodate and adapt to such variability, as evidenced by its strong performance under cross-layout and low-channel-count scenarios.

Table 11: Channels used in HEAR pretraining but not present in the HGD dataset.

| Unseen Channels in HGD Dataset |
| --- |
| AFF1, AFF2, AFF5h, AFF6h, AFp3h, AFp4h, CCP1h, CCP2h, CCP3h, CCP4h CCP5h, CCP6h, CPP1h, CPP2h, CPP3h, CPP4h, CPP5h, CPP6h, FCC1h, FCC2h FCC3h, FCC4h, FCC5h, FCC6h, FFC1h, FFC2h, FFC3h, FFC4h, FFC5h, FFC6h FFT7h, FFT8h, FTT7h, FTT8h, I1, I2, OI1h, OI2h, POO10h, POO3h POO4h, POO9h, PPO1, PPO10h, PPO2, PPO5h, PPO6h, PPO9h, TPP7h, TTP8h |

### G.2 CHANNEL CONFIGURATIONS IN CROSS-LAYOUT EXPERIMENT OF HGD DATASET

Table 11 lists the electrode channels that were included in the pretraining stage of the HEAR model but are entirely absent in the HGD dataset. These 50 channels, primarily consisting of high-density or custom-labeled electrodes (e.g., AFF*, AFp*, FCC*, CCP*, CPP*), reflect the use of a richer spatial configuration during pretraining based on extended 10-10 and 10-5 systems.

The absence of these channels in HGD indicates a non-trivial domain gap in spatial coverage between pretraining and downstream evaluation. Nevertheless, HEAR demonstrates strong generalization ability even when such channels are missing, suggesting that the model does not overfit to specific spatial topographies. This highlights the flexibility of HEAR's spatial inductive biases, which enable effective adaptation to lower-density configurations like those found in the HGD dataset.

This analysis underscores the practical value of designing EEG foundation models with robustness to heterogeneous electrode configurations—a key factor in ensuring transferability across real-world neurophysiological datasets.

## H ADDITIONAL VISUALIZATION

**Spatial Attention Analysis across Unseen Layouts.** We visualize the spatial distribution of attention in Figure 7, which depicts spatial attention scores projected onto the scalp topography for each unseen layout. A key observation is that despite large variations in electrode availability and positioning, the model consistently allocates high attention weights to semantically meaningful cortical regions. For instance, central-parietal areas (e.g., CPz, Cz, Pz) and motor-associated sites (e.g., C3, C4) exhibit strong activations across a wide range of layouts, including highly sparse ones such as Layouts 12 and 10.

Notably, the model retains its spatial inductive bias even under extreme channel sparsity. In configurations with fewer than 15 electrodes, attention remains concentrated in regions classically implicated in sensorimotor tasks, indicating a capacity for robust representation learning under severe spatial constraints. This consistent attentional focus suggests that the model implicitly infers spatial importance from its pretraining experience, transferring this knowledge to new geometries without explicit supervision. Moreover, layouts with highly lateralized channels (e.g., Layout 26 or 22) still preserve symmetric activation patterns, implying the model's ability to interpolate spatial structure from partial observations.

**Temporal Attention Analysis across Unseen Layouts.** To assess the spatial robustness of the proposed model, we conduct a cross-layout evaluation on the HGD dataset, in which training and evaluation layouts are explicitly disjoint. Figure 8 presents spatial attention activation for 18 unseen layouts, illustrating the model's ability to adapt under substantial sensor placement shifts. Despite the absence of overlapping channel configurations between training and test layouts, we observe consistent and structured patterns of attention across epochs, particularly in midline and parietal regions (e.g., Pz, CPz), which are likely to encode transferable neural features. Notably, layouts such as 18, 16, and 12 show heightened activation in similar spatial clusters across time, suggesting the model effectively recalibrates its spatial priors to align with semantically relevant regions even when raw electrode locations change. This robustness is critical for real-world BCI deployment, where electrode positioning may vary across sessions and subjects. The results affirm the proposed model's capacity to generalize beyond seen sensor geometries, capturing invariant spatial representations under configuration shifts. Additional analysis of the unseen channel identities is provided in Appendix 11.

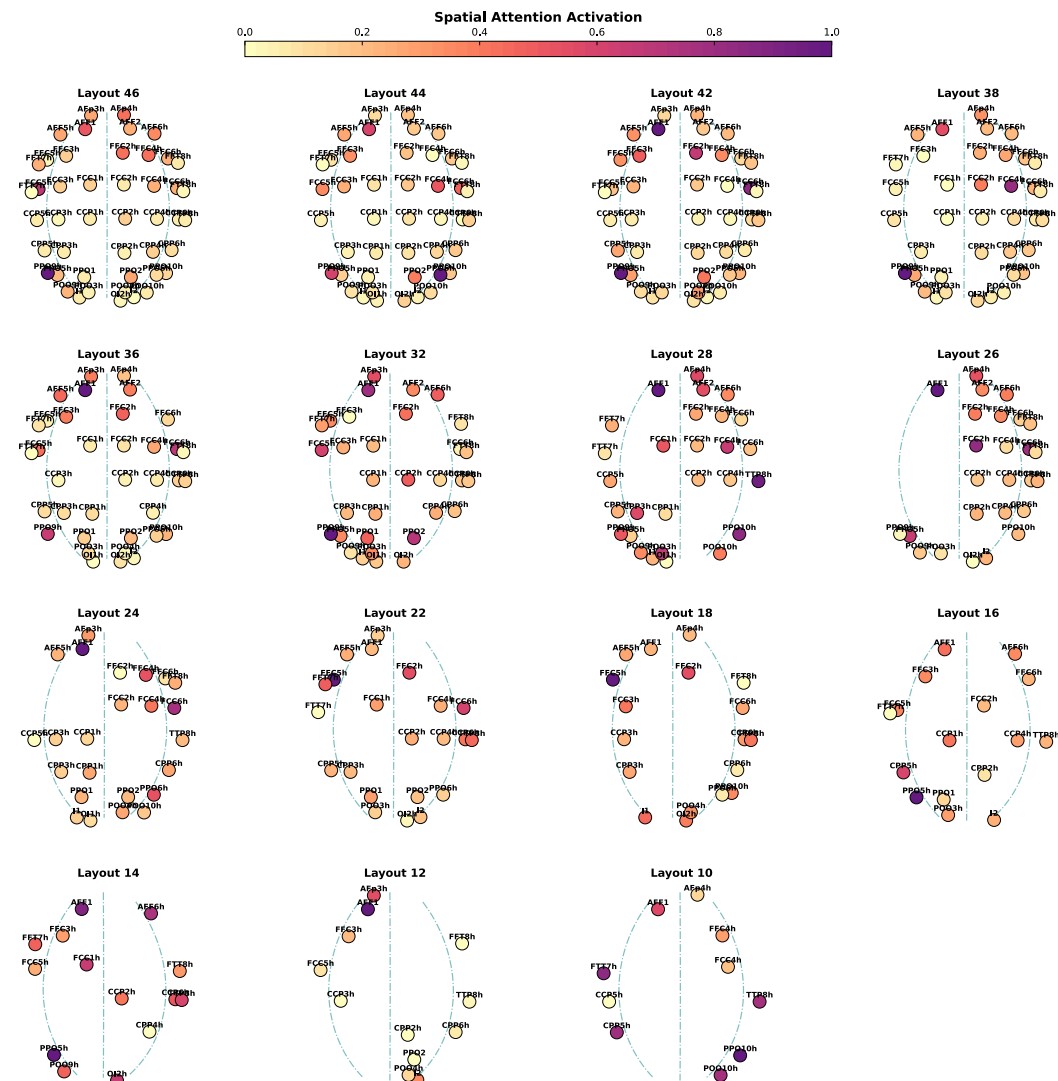

Figure 7: Channel activation visualization of different channel configurations.

Together, these results affirm that the proposed attention mechanism not only adapts temporally across unseen configurations but also preserves meaningful spatial activation priors, demonstrating its viability for real-world deployment scenarios where electrode placement may vary widely across sessions or subjects.

## I    DATA PROCESSING APPROACH

**Data Preprocessing:** To ensure consistency across all datasets during both pretraining and fine-tuning, we adopted a standardized EEG preprocessing pipeline. (1) We first applied average channel referencing, subtracting the mean signal across all electrodes from each channel to reduce common-mode artifacts. (2) All signals were then resampled to 200 Hz to normalize temporal resolution and reduce computational cost. (3) Finally, a bandpass FIR filter (1–75 Hz, zero-phase, Hamming window) was applied to remove low-frequency drifts and high-frequency noise, with the upper cutoff limited to the minimum of 75 Hz or the Nyquist frequency Srinivasan et al. (1998).

**Data Segmentation:** For all downstream datasets listed in Table 2, we standardized the segmentation and splitting procedures by utilizing task-specific event timestamps provided within each dataset.

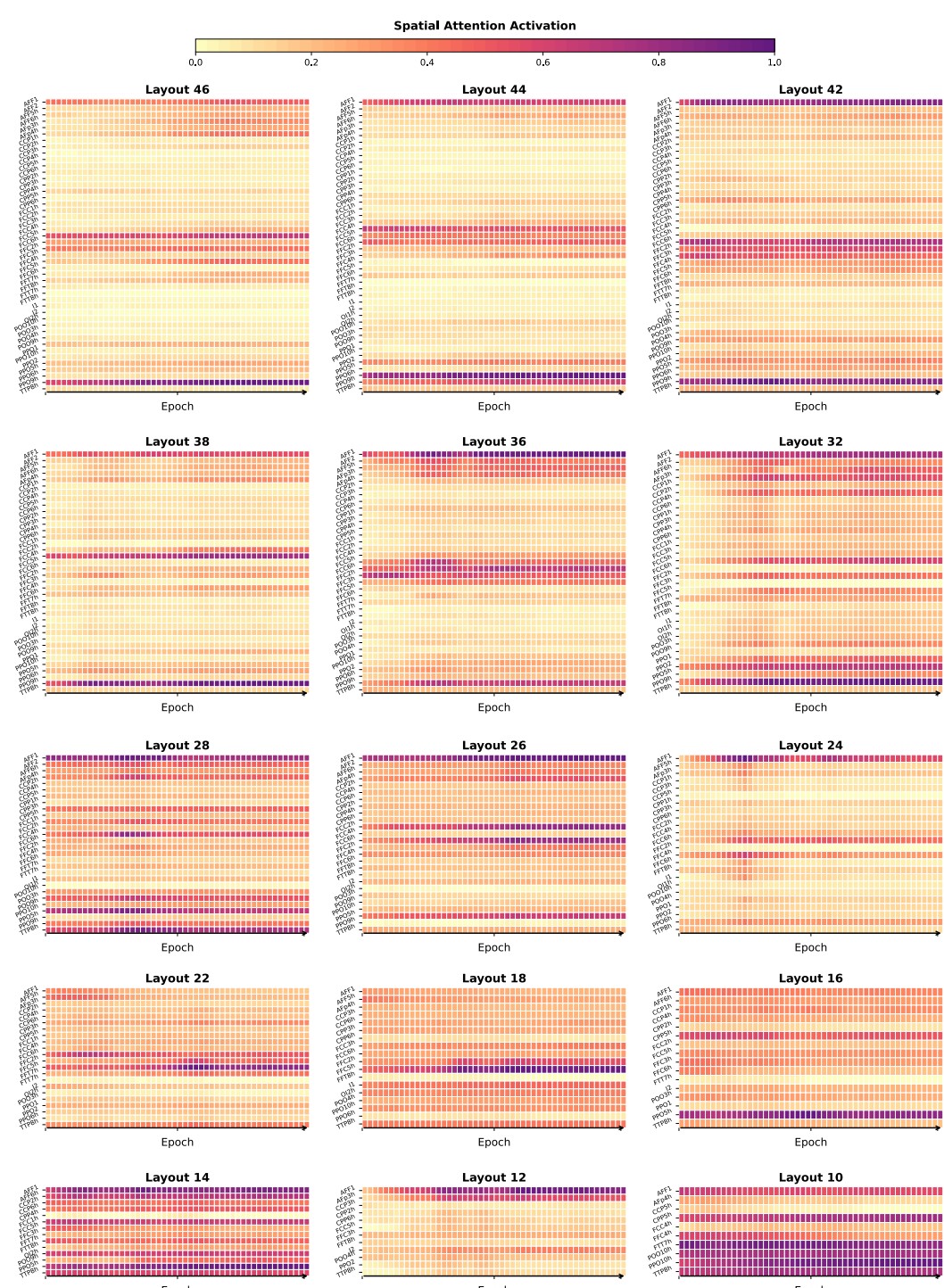

Figure 8: Channel activation heatmap of cross-layout experiments.

# J DESCRIPTION OF THE PUBLIC DATASETS UTILIZED IN HEAR

The following section provides a detailed description of the datasets listed in Table 2, offering a comprehensive understanding of their characteristics.

**BCI-IV-2A (Available: bbci.de/competition/iv/download/):** A dataset from the **BCI Competition 2008 - Graz data set A** that includes EEG recordings from 9 subjects performing four motor imagery tasks: imagining movements of the left hand, right hand, both feet, and tongue. Each subject completed two sessions, with each session consisting of 6 runs and 288 trials (12 trials per class per run). The paradigm involved a fixation cross and an auditory cue prompting the imagery task, which lasted until the cross disappeared at 6 seconds. EEG data were recorded using 22 Ag/AgCl electrodes (10-20 system) at 250 Hz, bandpass filtered between 0.5 Hz and 100 Hz, with three additional EOG channels for artifact processing. The dataset is stored in GDF format, with separate training and evaluation files for each subject. The training data includes class labels, while the evaluation data is used for testing.

**(HGD) High Gamma Dataset Schirrmeister et al. (2017):** The **High-Gamma Dataset (HGD)** is a large-scale EEG dataset designed for motor imagery decoding, featuring recordings from 14 healthy subjects performing imagined and executed movements of the left hand, right hand, feet, and rest. EEG signals were recorded using a 64-channel cap with the 10-10 electrode system at a sampling rate of 500 Hz, bandpass filtered between 0.1–125 Hz to emphasize high-gamma activity. Each trial spans 4 seconds, starting 500 ms before the cue onset to capture preparatory neural activity. The dataset is structured to support both trial-wise and cropped training strategies, with cropped training leveraging sliding time windows to enhance decoding performance.

**OpenBMI Lee et al. (2019):** The **OpenBMI** Dataset is a large-scale EEG dataset designed to facilitate brain-computer interface (BCI) research across multiple paradigms. It includes recordings from a large number of subjects over multiple sessions, covering motor imagery, event-related potential (ERP), and steady-state visually evoked potential paradigms. EEG signals were recorded using a high-density electrode system with a standardized montage, ensuring broad applicability in neural decoding tasks. In addition to paradigm-specific tasks, the dataset includes resting-state recordings, artifacts, and electromyographic signals from both arms. Moreover, psychological and physiological data from participants were collected through structured questionnaires to explore inter-subject variability in BCI control performance. These comprehensive recordings enable systematic evaluations of decoding accuracy, cross-subject/session variability, and BCI illiteracy across paradigms, making OpenBMI a valuable benchmark for EEG-based BCI research.

**(EEGMAT) EEG During Mental Arithmetic Tasks Zyma et al. (2019):** The **EEGMAT** Dataset is an EEG dataset designed to investigate brain activity under cognitive workload, specifically during mental arithmetic tasks (serial subtraction). EEG signals were recorded from 36 healthy subjects performing arithmetic calculations, with a separate resting-state recording for comparison. The dataset was collected using a 23-channel Neurocom EEG system, following the 10-20 electrode placement system, with a sampling rate of 500 Hz and bandpass filtering between 0.5–45 Hz. Subjects were divided into good counters and bad counters based on their task performance, enabling the study of inter-individual differences in cognitive load processing. The recordings include artifact-free EEG segments for both conditions, supporting analysis through power spectral density, coherence, and nonlinear signal processing techniques.

**(TUEP) TUH EEG Epilepsy dataset Veloso et al. (2017):** The **TUH EEG Epilepsy dataset** is a specialized subset of the TUH EEG dataset, designed to facilitate automatic EEG analysis for epilepsy detection. It consists of 570 sessions from 200 patients, categorized into two groups based on clinical history, medications, and EEG features indicative of epilepsy. The dataset contains 1,799 EEG recordings in European Data Format (EDF), along with corresponding neurologist reports. Among these, 1,473 files from 436 sessions (100 patients) belong to the epilepsy group, while 326 files from 134 sessions (100 patients) belong to the non-epilepsy group. This dataset serves as a valuable resource for developing and benchmarking machine learning algorithms for epilepsy classification and seizure detection.

**(HMCSleep) Haaglanden Medisch Centrum Sleep Center Database Alvarez-Estevez & Rijsman (2021):** The **HMC Sleep Dataset** is a publicly available polysomnographic (PSG) dataset collected from the Haaglanden Medisch Centrum Sleep Center (HMC), Netherlands, to facilitate research in automatic sleep staging and sleep disorder analysis. The dataset consists of 154 full-night PSG recordings, acquired from a heterogeneous population of patients referred for clinical sleep examination in 2018. Data collection followed standard clinical procedures, ensuring real-world variability in sleep disorder diagnosis. Each PSG recording includes EEG, electromyography (EMG), and electrooculography (EOG) signals, resampled at 100 Hz to maintain signal integrity while

Table 12: Overview of HEAR Dataset.

Datasets with multiple channel layouts are divided into sub-datasets based on channel categorization.

Datasets with dual EEG channels, where the first channel in the lead is selected as input.

| | Datasets | Hours | Channel Configuration | Available Channel Number | Targets |
|---|---|---|---|---|---|
| **Pretraining** | OpenBMI | 91.6 | 62 EEG Channels | 62 (62) | - |
| | Migrainedb | 21.2 | 133 EEG Channels, 1 ECG Channel, 10 Unknowns | 133 (78) | - |
| | PhysioNetP300 | 2.27 | 64 EEG Channels, 6 Unknowns | 64 | - |
| | EEGMAT | 2.4 | 20 EEG Channels, 1 ECG Channel | 20 (19) | - |
| | KaggleERN | 30.0 | 56 EEG Channels, 2 Unknowns | 56 (55) | - |
| | TUEP | 631.8 | **34 Channel Configurations:** 17, 18, 25, 27, 28, 29, 30, 31, 32, 33, 34, 35, 36, 41 | 17 (17), 27 (23), 29 (23), 28 (23), 30 (23), 31 (21), 33 (24), 31 (23), 32 (23), 33 (23), 34 (23), 35 (23), 36 (23), 31 (23) | - |
| | TUEV | 148.7 | **14 Channel Configurations:** 24, 25, 27, 28, 30, 31, 32, 33 | 24 (22), 24 (23), 25 (23), 27 (23), 28 (23), 30 (23), 31 (23), 32 (23), 33 (23), 25 (23) | - |
| | HMCSleep | 582.5 | **8 Dual EEG Channels** | 4 Using the first lead-channel | - |

| | Datasets | Hours | Channel Configuration | Available Channel Number | Targets |
|---|---|---|---|---|---|
| **Downstream** | BCI-IV-1 | 3.7 | 59 EEG Channels | 59 (49) | 2 |
| | BCI-IV-2B | 26.3 | 3 EEG Channels, 3 EoG Channels | 3 (3) | 2 |
| | EEGMMIDB | 48.5 | 64 EEG Channels | 64 | 4 |
| | LargeMI | 59.4 | 23 EEG Channels | 22 | 4 |
| | SHUDB | 12.4 | 32 EEG Channels | 32 | 2 |
| | HGD | 28.7 | 133 EEG Channels | 133 (78) | 2 |
| | BCI-IV-2A | 13.4 | 22 EEG Channels, 3 EoG Channels | 22 (22) | 4 |

| | Datasets | Hours | Channel Configuration | Available Channel Number | Targets |
|---|---|---|---|---|---|
| **Aux.** | SleepEDFx | 3849.0 | **2 Dual Channel Configurations:** 5 Dual EEG Channels, 7 Dual EEG Channels | 4 Using the first lead-channel | - |
| | TUAB | 1137.3 | **17 Channel Configurations:** 27, 28, 29, 30, 31, 33, 34, 35, 36 | 27 (22), 28 (23), 29 (23), 30 (23), 31 (23), 33 (23), 33 (24), 34 (23), 35 (23), 36 (23) | - |
| | CHB-MIT | 1060.9 | **12 Dual Channel Configurations:** 22, 23, 24, 25, 28, 29, 31, 38 | 10 Using the first lead-channel | - |
| | TUSL | 27.6 | **12 Channel Configurations:** 27, 28, 32, 33, 34, 36, 41 | 27 (23), 28 (23), 30 (23), 32 (23), 33 (23), 33 (24), 34 (23), 36 (23), 41 (22) | - |
| | CAPSleep | 1004.5 | **50 Dual Channel Configurations:** 5, 8, 9, 10, 11, 12, 13, 14, 15, 16, 17, 18, 20, 21, 22, 23, 24, 27, 34, 36 | 5 Using the first lead-channel | - |

optimizing data processing. Standard 30-second epoch-based scoring was applied following AASM guidelines, classifying sleep stages into wakefulness, N1, N2, N3, and REM sleep. The dataset is fully anonymized and provides a valuable benchmark for deep learning-based sleep staging models, supporting generalization across diverse sleep databases.

**(CAPSleep) CAP Sleep Database Terzano et al. (2001):** The **CAP Sleep Database** is a polysomnographic (PSG) dataset collected at the Sleep Disorders Center of Ospedale Maggiore, Parma, Italy, designed for the study of Cyclic Alternating Pattern (CAP) during sleep. The dataset comprises 108 overnight PSG recordings, including 16 healthy subjects and 92 pathological recordings covering nocturnal frontal lobe epilepsy (NFLE, 40), REM behavior disorder (RBD, 22), periodic leg movements (PLM, 10), insomnia (9), narcolepsy (5), sleep-disordered breathing (SDB, 4), and bruxism (2).

Each recording contains at least three EEG channels (F3/F4, C3/C4, O1/O2) referenced to A1/A2, along with EOG (2 channels), EMG (submental and bilateral anterior tibial), respiration signals (airflow, thoracic and abdominal effort), and EKG. Additional bipolar EEG derivations (e.g., Fp1-F3, F3-C3, C3-P3, P3-O1) are provided according to the 10-20 international system. The dataset serves as a valuable resource for automatic CAP detection, sleep disorder characterization, and quantitative analysis of sleep instability, supporting both clinical and computational research in sleep medicine.

**(CHB-MIT) CHB-MIT Scalp EEG Database Shoeb (2009):** The **CHB-MIT Scalp EEG Database** is a publicly available dataset designed for the study of epileptic seizure detection and prediction. It consists of 844 hours of continuous scalp EEG recordings collected from 23 pediatric epilepsy patients at Children's Hospital Boston (CHB). The recordings include 163 test seizures, with a median detection delay of 3 seconds, enabling the development of real-time seizure onset detection algorithms. EEG signals were recorded using a scalp electrode montage, capturing a wide range of seizure patterns across different brain regions. The dataset serves as a benchmark for machine learning-based seizure detection, supporting research in automated diagnosis, patient-specific seizure prediction, and neurostimulation-based intervention strategies.

**(BCI-IV-1) BCI Competition IV Dataset 1 Blankertz et al. (2007):** The **BCI Competition IV Dataset 1** is a publicly available EEG dataset designed for motor imagery-based BCI research. The dataset was collected using the Berlin Brain-Computer Interface (BBCI) system, which leverages machine-learning techniques to extract subject-specific sensorimotor patterns for rapid BCI calibration. EEG signals were recorded from 10 subjects, who performed imagined left-hand, right-hand, and foot movements, with data acquisition carried out using a 128-channel EEG system. The recordings were bandpass filtered between 0.05–200 Hz and downsampled to 100 Hz, ensuring high signal quality for classification tasks. Unlike traditional BCI approaches requiring extensive user training, the dataset emphasizes short calibration sessions (20 minutes) followed by machine learning-based adaptation, enabling fast and effective BCI control. The dataset serves as a benchmark for feature extraction, classification, and real-time BCI applications.

**(BCI-IV-2B) BCI Competition IV Dataset 2B (Available: bbci.de/competition/iv/download/):** The BCI Competition IV Dataset 2b (BCI IV-2b) is an EEG dataset designed for motor imagery-based brain-computer interface (BCI) research. It consists of EEG recordings from 9 right-handed subjects, each participating in five sessions recorded on different days. The dataset includes three bipolar EEG channels (C3, Cz, and C4) sampled at 250 Hz, bandpass-filtered between 0.5–100 Hz, with a 50 Hz notch filter applied. The experimental paradigm involves cue-based motor imagery (MI) tasks, where subjects imagine left-hand and right-hand movements. The first two sessions contain training data without feedback, while the last three sessions include real-time feedback using a smiley-based visual cue. Additionally, EOG signals were recorded to facilitate artifact removal. The dataset is provided in General Data Format (GDF) and serves as a benchmark for motor imagery classification, feature extraction, and adaptive BCI systems.

**(LargeMI) Large Electroencephalographic Motor Imagery Dataset Kaya et al. (2018):** The **Large Electroencephalographic Motor Imagery Dataset (LargeMI)** is a comprehensive EEG dataset specifically designed for advancing research in electroencephalographic brain-computer interfaces (BCI). This large-scale dataset aggregates approximately 60 hours of EEG recordings from 75 experiments conducted with 13 participants, encompassing a total of about 60,000 trials of mental imagery tasks. It features recordings from four different BCI interaction paradigms, which include up to six distinct interaction states for motor imagery. With an average of 4.8 hours of EEG data and 4,600 mental imagery examples per participant, the dataset is notable for its extensive longitudinal span, broad lateral coverage, and significant interaction complexity. It serves as a substantial benchmark for developing and validating machine learning models in areas such as motor imagery classification, BCI robustness testing, and longitudinal neural signal analysis.

**(KaggleERN) Kaggle ERN Dataset Margaux et al. (2012):** The **Kaggle ERN Dataset** is an EEG dataset designed for the study of error-related negativity (ERN) and feedback-related negativity (FRN), which are key components in error monitoring and brain-computer interface (BCI) applications. The dataset was collected using a P300-based speller paradigm, where 16 healthy subjects were tasked with selecting letters from a matrix while their brain responses to correct and erroneous feedback were recorded. EEG signals were acquired from 32 Ag/AgCl electrodes, following the extended 10-20 system, with a 600 Hz sampling rate, and referenced to the nose. The dataset enables the study of real-time error detection and correction, as it includes both subjective user feedback and

neurophysiological error potentials (ErrP). It serves as a benchmark for machine learning models in EEG-based error correction, human-computer interaction, and adaptive BCI systems.

**(EEGMMIDB) EEG Motor Movement/Imagery Dataset Schalk et al. (2004):** The **EEG Motor Movement/Imagery Dataset** is a widely used EEG dataset designed for motor imagery and movement-related BCI research. The dataset consists of EEG recordings from 109 subjects, who performed motor execution and motor imagery tasks, including left/right fist clenching and foot movements. EEG signals were recorded using a 64-channel EEG system following the 10-10 electrode placement system, with a 160 Hz sampling rate. Each subject participated in two experimental runs, one for actual movement and one for motor imagery, enabling direct comparisons between executed and imagined movements. The dataset serves as a benchmark for developing EEG-based movement classification algorithms, motor imagery decoding, and real-time BCI applications.

**(PhysioNetP300) PhysioNet P300 Dataset Citi et al. (2010):** The **PhysioNet P300 Dataset** is an EEG dataset designed for research on P300-based BCIs, particularly in the context of Donchin's P300 speller paradigm. The dataset includes EEG recordings from subjects engaged in a P300 spelling task, where rare target stimuli elicit a P300 event-related potential (ERP), enabling the selection of characters. EEG signals were recorded using a multichannel electrode system, with a focus on detecting P300 amplitude variations influenced by stimulus sequence and target delays. The dataset facilitates the development of machine learning models for ERP detection, BCI accuracy optimization, and user-specific adaptation strategies, making it a valuable resource for advancing P300-based BCI applications.

**(SleepEDFx) Sleep-EDF Expanded Dataset Kemp et al. (2000):** The **Sleep-EDF Expanded Dataset** is a publicly available PSG dataset designed for sleep research, including automatic sleep staging, sleep disorder analysis, and circadian rhythm studies. It consists of multiple nights of PSG recordings from healthy subjects and patients with sleep disorders, collected using standardized sleep monitoring protocols. EEG signals were recorded from multiple scalp electrodes, along with EOG, EMG, ECG, and respiratory signals, enabling comprehensive sleep pattern analysis. The dataset includes manual sleep stage annotations following AASM or R&K scoring criteria, classifying epochs into wake, N1, N2, N3, and REM sleep stages. With its high-quality signals and diverse subject pool, Sleep-EDFx serves as a benchmark for machine learning-based sleep stage classification, sleep disorder detection, and longitudinal sleep studies.

**(TUAB) Temple University Hospital Abnormal EEG dataset Obeid & Picone (2016):** The Temple University Hospital Abnormal EEG dataset is a large-scale, clinically sourced EEG dataset designed for machine learning-based EEG analysis. It consists of 16,986 EEG sessions from 10,874 unique subjects, spanning a diverse population with ages ranging from infants to elderly individuals. The recordings were obtained from clinical EEG exams conducted at Temple University Hospital (TUH) over 14 years, ensuring real-world variability in electrode placement, recording conditions, and patient states. EEG data were collected using multiple channel configurations, with most recordings containing 31 EEG channels, alongside supplementary EKG, EMG, and photic stimuli channels. Sampling rates primarily include 250 Hz, 256 Hz, 400 Hz, and 512 Hz. The dataset is de-identified and paired with neurologist reports, making it a valuable resource for seizure detection, epilepsy classification, and general EEG-based diagnostic studies.

**(TUSL) TUH EEG Slowing dataset Veloso et al. (2017):** The **TUH EEG Slowing dataset** is a specialized subset of the Temple University Hospital EEG dataset Veloso et al. (2017), designed for the study of EEG slowing events and their differentiation from seizure activity. The dataset consists of 38 unique patients, 75 EEG sessions, and 112 aggregated files, with 300 annotated events of seizures, independent slowing events, and complex background events, each lasting 10 seconds. The EEG data follows a term-based annotation approach, ensuring event-level labeling across all channels, making it highly suitable for machine learning-based detection and classification of EEG slowing patterns. TUSL serves as a valuable resource for automated EEG interpretation, seizure differentiation, and clinical neuroscience research.

## K    DESCRIPTION OF CONSTRUCTED HETEROGENEOUS *HEAR Dataset*

To enable heterogeneous training in EEG modeling, we introduce the HEAR Dataset, a structured collection of EEG recordings that preserves the full diversity of electrode configurations. Unlike

Table 13: HEAR Dataset with single configuration.

| Dataset Name | File with Single Configuration | Channel Name |
|---|---|---|
| *BCIC-IV-2A* | 25-3afd.hdf5 | Fz, FC3, FC1, FCz, FC2, FC4, C5, C3, C1, Cz, C2, C4, C6, CP3, CP1, CPz, CP2, CP4, P1, POz, Pz, P2, EOG-left, EOG-central, EOG-right |
| *HGD* | 133-4f38.hdf5 | EEG M1, EEG OI2h, EEG Fp1, EEG Fp2, EEG Fpz, EEG PPO6h, EEG PPO10h, EEG POO9h, EEG POO3h, EEG POO4h, EEG POO10h, EEG OI1h, EEG F3, EEG T7, EEG C3, EEG Cz, EEG C4, EEG T8, EEG M2, EEG CP5, EEG CP1, EEG CP2, EEG CP6, EEG P7, EEG P3, EEG Pz, EEG P4, EEG P8, EEG POz, EEG O1, EEG Oz, EEG O2, EOG EOGh, EOG EOGv, EMG EMG_RH, EMG EMG_LH, EMG EMG_RF, EEG AF7, EEG AF3, EEG AF4, EEG AF8, EEG F5, EEG F1, EEG F2, EEG F6, EEG FC3, EEG FCz, EEG FC4, EEG C5, EEG C1, EEG C2, EEG C6, EEG CP3, EEG CPz, EEG CP4, EEG P5, EEG P1, EEG P2, EEG P6, EEG PO5, EEG PO3, EEG PO4, EEG PO6, EEG FT7, EEG FT8, EEG TP7, EEG TP8, EEG PO7, EEG PO8, EEG FT9, EEG FT10, EEG TPP9h, EEG TPP10h, EEG PO9, EEG PO10, EEG P9, EEG P10, EEG AFF1, EEG AFz, EEG AFF2, EEG FFC5h, EEG FFC3h, EEG FFC4h, EEG FFC6h, EEG FCC5h, EEG FCC3h, EEG FCC4h, EEG FCC6h, EEG CCP5h, EEG CCP3h, EEG CCP4h, EEG CCP6h, EEG CPP5h, EEG CPP3h, EEG CPP4h, EEG CPP6h, EEG PPO1, EEG PPO2, EEG I1, EEG Iz, EEG I2, EEG AFp3h, EEG AFp4h, EEG AFF5h, EEG AFF6h, EEG FFT7h, EEG FFC1h, EEG FFC2h, EEG FFT8h, EEG FTT9h, EEG FTT7h, EEG FCC1h, EEG FCC2h, EEG FTT8h, EEG FTT10h, EEG TTP7h, EEG CCP1h, EEG Fz, EEG F4, EEG F8, EEG FC5, EEG FC1, EEG FC2, EEG FC6, EEG F7, EEG CCP2h, EEG TTP8h, EEG TPP7h, EEG CPP1h, EEG CPP2h, EEG TPP8h, EEG PPO9h, EEG PPO5h |
| *OpenBMI* | 62-b361.hdf5 | GSR1, GSR2, Erg1, Erg2, Resp, Plet, Temp, Status, F5, F3, F1, FFT9h, FFT7h, FFC5h, FFC3h, FFC1h, FT9, FT7, FC5, FC3, FC1, FTT9h, FTT7h, FCC5h, FCC3h, FCC1h, T7, C5, C3, C1, TTP7h, CCP5h, CCP3h, CCP1h, TP9, TP7, CP5, CP3, CP1, CPz, TPP7h, CPP5h, CPP3h, CPP1h, P9, P7, P5, P3, P1, Pz, PPO9h, PPO5h, PPO1h, PO7, PO3, POz, PO9, POO9h, O1, POO1, I1, OI1h, Oz, Iz, Fpz, Fp2, AFp2, AFz, AF4, AF8, AFF2h, AFF6h, Fz, F2, F4, F6, F8, F10, FFC2h, FFC4h, FFC6h, FFT8h, FFT10h, FCz, FC2, FC4, FC6, FT8, FT10, FCC2h, FCC4h, FCC6h, FTT8h, FTT10h, Cz, C2, C4, C6, T8, CCP2h, CCP4h, CCP6h, TTP8h, CP2, CP4, CP6, TP8, TP10, CPP2h, CPP4h, CPP6h, TPP8h, P2, P4, P6, P8, P10, PPO2h, PPO6h, PPO10h, PO4, PO8, PO10, POO2, O2, POO10h, OI2h, I2, Fp1, AFp1, AF7, AF3, AFF5h, AFF1h, F9, F7, M1, M2, LO1, LO2, IO1, SO1, IO2, ECG |
| *EEGMat* | 21-6e05.hdf5 | EEG Fp1, EEG Fp2, EEG F3, EEG F4, EEG F7, EEG F8, EEG T3, EEG T4, EEG C3, EEG C4, EEG T5, EEG T6, EEG P3, EEG P4, EEG O1, EEG O2, EEG Fz, EEG Cz, EEG Pz, EEG A2-A1, ECG ECG |
| *HMCSleep* | 8-8a48.hdf5 | EEG F4-M1, EEG C4-M1, EEG O2-M1, EEG C3-M2, EMG chin, EOG E1-M2, EOG E2-M2, ECG |
| *BCIC-IV-1* | 59-f534.hdf5 | AF3, AF4, F5, F3, F1, Fz, F2, F4, F6, FC5, FC3, FC1, FCz, FC2, FC4, FC6, CFC7, CFC5, CFC3, CFC1, CFC2, CFC4, CFC6, CFC8, T7, C5, C3, C1, Cz, C2, C4, C6, T8, CCP7, CCP5, CCP3, CCP1, CCP2, CCP4, CCP6, CCP8, CP5, CP3, CP1, CPz, CP2, CP4, CP6, P5, P3, P1, Pz, P2, P4, P6, PO1, PO2, O1, O2 |
| *BCIC-IV-2B* | 6-835e.hdf5 | EEG:C3, EEG:Cz, EEG:C4, EOG:ch01, EOG:ch02, EOG:ch03 |
| *KaggleERN* | 58-0ed0.hdf5 | Fp1, Fp2, AF7, AF3, AF4, AF8, F7, F5, F3, F1, Fz, F2, F4, F6, F8, FT7, FC5, FC3, FC1, FCz, FC2, FC4, FC6, FT8, T7, C5, C3, C1, Cz, C2, C4, C6, T8, TP7, CP5, CP3, CP1, CPz, CP2, CP4, CP6, TP8, P7, P5, P3, P1, Pz, P2, P4, P6, P8, PO7, POz, PO8, O1, O2, EOG, FeedBackEvent |
| *Migrainedb* | 144-d615.hdf5 | GSR1, GSR2, Erg1, Erg2, Resp, Plet, Temp, Status, F5, F3, F1, FFT9h, FFT7h, FFC5h, FFC3h, FFC1h, FT9, FT7, FC5, FC3, FC1, FTT9h, FTT7h, FCC5h, FCC3h, FCC1h, T7, C5, C3, C1, TTP7h, CCP5h, CCP3h, CCP1h, TP9, TP7, CP5, CP3, CP1, CPz, TPP7h, CPP5h, CPP3h, CPP1h, P9, P7, P5, P3, P1, Pz, PPO9h, PPO5h, PPO1h, PO7, PO3, POz, PO9, POO9h, O1, POO1, I1, OI1h, Oz, Iz, Fpz, Fp2, AFp2, AFz, AF4, AF8, AFF2h, AFF6h, Fz, F2, F4, F6, F8, F10, FFC2h, FFC4h, FFC6h, FFT8h, FFT10h, FCz, FC2, FC4, FC6, FT8, FT10, FCC2h, FCC4h, FCC6h, FTT8h, FTT10h, Cz, C2, C4, C6, T8, CCP2h, CCP4h, CCP6h, TTP8h, CP2, CP4, CP6, TP8, TP10, CPP2h, CPP4h, CPP6h, TPP8h, P2, P4, P6, P8, P10, PPO2h, PPO6h, PPO10h, PO4, PO8, PO10, POO2, O2, POO10h, OI2h, I2, Fp1, AFp1, AF7, AF3, AFF5h, AFF1h, F9, F7, M1, M2, LO1, LO2, IO1, SO1, IO2, ECG |
| *EEGMMIDB* | 64-b33f.hdf5 | Fc5., Fc3., Fc1., Fcz., Fc2., Fc4., Fc6., C5.., C3.., C1.., Cz.., C2.., C4.., C6.., Cp5., Cp3., Cp1., Cpz., Cp2., Cp4., Cp6., Fp1., Fpz., Fp2., Af7., Af3., Afz., Af4., Af8., F7.., F5.., F3.., F1.., Fz.., F2.., F4.., F6.., F8.., Ft7., Ft8., T7.., T8.., T9.., T10., Tp7., Tp8., P7.., P5.., P3.., P1.., Pz.., P2.., P4.., P6.., P8.., Po7., Po3., Poz., Po4., Po8., O1.., Oz.., O2.., Iz.. |

conventional approaches that standardize data by selecting only common channels, our dataset retains all original channel information and organizes recordings into subsets based on their specific electrode configurations. Each subset is stored separately in an HDF5 file, ensuring that no spatial information is lost while facilitating efficient heterogeneous processing and channel modeling. This structure allows models to learn from a wide range of EEG setups, promoting robustness across different hardware and experimental designs. Table 13 and Table 14 summarize datasets with homogeneous

Table 14: HEAR Dataset with heterogeneous configurations (P1).

| Dataset Name | File with Heterogeneous Configurations | Channel Proportions among all Configurations |
|---|---|---|
| *TUEP* | **(34 Channel Configurations)** 17-e44c.hdf5, 18-1435.hdf5, 27-2288.hdf5, 28-8134.hdf5, 29-6d29.hdf5, 29-a3db.hdf5, 29-8395.hdf5, 30-b5ad.hdf5, 30-8cc3.hdf5, 30-1ba1.hdf5, 31-4aba.hdf5, 31-f727.hdf5, 31-384a.hdf5, 31-8a53.hdf5, 31-11eb.hdf5, 32-4f6a.hdf5, 32-1203.hdf5, 32-893d.hdf5, 32-9956.hdf5, 32-dfe0.hdf5, 33-b7ae.hdf5, 33-08ac.hdf5, 33-17fd.hdf5, 33-aec0.hdf5, 34-fcd3.hdf5, 34-8b7a.hdf5, 34-e2d2.hdf5, 34-56c4.hdf5, 34-2300.hdf5, 35-2d75.hdf5, 35-cc48.hdf5, 35-398b.hdf5, 36-8e4e.hdf5, 41-7f15.hdf5 | EEG F7-REF (93.60%), EEG T6-REF (93.60%), EEG C3-REF (93.60%), EEG F4-REF (93.60%), EEG C4-REF (93.60%), EEG O1-REF (93.60%), EEG P4-REF (93.60%), EEG FP1-REF (93.60%), EEG FP2-REF (93.60%), EEG O2-REF (93.60%), EEG T3-REF (93.60%), EEG T4-REF (93.60%), EEG F8-REF (93.60%), EEG CZ-REF (93.60%), EEG P3-REF (93.60%), EEG T5-REF (93.60%), EEG F3-REF (93.60%), EEG FZ-REF (93.26%), EEG PZ-REF (93.26%), EEG EKG1-REF (90.34%), BURSTS (90.08%), SUPPR (90.08%), IBI (90.08%), EEG T1-REF (90.08%), EEG T2-REF (90.08%), EEG A2-REF (81.46%), EEG A1-REF (81.46%), EMG-REF (53.44%), EEG C4P-REF (47.30%), EEG C3P-REF (47.30%), EEG SP1-REF (46.87%), EEG SP2-REF (46.82%), EEG 31-REF (38.64%), EEG 32-REF (38.60%), PHOTIC-REF (36.38%), EEG LOC-REF (28.76%), EEG ROC-REF (28.76%), EEG 30-REF (25.46%), EEG 29-REF (25.46%), EEG 26-REF (17.01%), EEG 27-REF (16.88%), EEG 28-REF (16.88%), EEG FP1-LE (6.40%), EEG F4-LE (6.40%), EEG F3-LE (6.40%), EEG FP2-LE (6.40%), EEG EKG-LE (6.40%), EEG A2-LE (6.40%), EEG A1-LE (6.40%), EEG C4-LE (6.40%), EEG C3-LE (6.40%), EEG CZ-LE (6.40%), EEG OZ-LE (6.40%), EEG P4-LE (6.40%), EEG P3-LE (6.40%), EEG O1-LE (6.40%), EEG O2-LE (6.40%), EEG F8-LE (6.40%), EEG PZ-LE (6.40%), EEG T6-LE (6.40%), EEG T3-LE (6.40%), EEG F7-LE (6.40%), EEG T4-LE (6.40%), EEG FZ-LE (6.40%), EEG T5-LE (6.40%), EEG 30-LE (6.40%), EEG 28-LE (6.22%), EEG 29-LE (6.22%), EEG 27-LE (5.96%), EEG 26-LE (5.96%), PHOTIC PH (5.92%), EEG PG2-LE (5.92%), EEG PG1-LE (5.92%), EEG 32-LE (5.48%), EEG 31-LE (5.48%), DC4-DC (5.48%), DC5-DC (5.48%), DC6-DC (5.48%), DC8-DC (5.48%), DC2-DC (5.48%), DC1-DC (5.48%), DC7-DC (5.48%), DC3-DC (5.48%), EEG 25-REF (3.22%), EEG 20-REF (3.18%), EEG 23-REF (3.18%), EEG 21-REF (3.18%), EEG 22-REF (3.18%), EEG 24-REF (3.18%), EEG T2-LE (0.91%), EEG T1-LE (0.91%), EEG PG2-REF (0.65%), EEG PG1-REF (0.65%), EEG 24-LE (0.48%), EEG 23-LE (0.48%), EEG SP1-LE (0.44%), EEG SP2-LE (0.44%), EEG LUC-LE (0.17%), EEG RLC-LE (0.17%) |
| *TUEV* | **(14 Channel Configurations)** 24-0497.hdf5, 24-ceb3.hdf5, 25-f934.hdf5, 25-883d.hdf5, 27-4dc7.hdf5, 28-3bdb.hdf5, 28-0186.hdf5, 30-d15e.hdf5, 31-74f9.hdf5, 31-e19b.hdf5, 32-e7d5.hdf5, 32-3a05.hdf5, 32-2f5f.hdf5, 33-cfc2.hdf5 | EEG FP1-REF (100.0%), EEG FP2-REF (100.0%), EEG F3-REF (100.0%), EEG F4-REF (100.0%), EEG C3-REF (100.0%), EEG C4-REF (100.0%), EEG P3-REF (100.0%), EEG P4-REF (100.0%), EEG O1-REF (100.0%), EEG O2-REF (100.0%), EEG F7-REF (100.0%), EEG F8-REF (100.0%), EEG T3-REF (100.0%), EEG T4-REF (100.0%), EEG T5-REF (100.0%), EEG T6-REF (100.0%), EEG A1-REF (100.0%), EEG A2-REF (100.0%), EEG FZ-REF (100.0%), EEG CZ-REF (100.0%), EEG PZ-REF (100.0%), EEG T1-REF (99.81%), EEG T2-REF (99.81%), EEG EKG1-REF (96.53%), PHOTIC-REF (75.68%), EEG ROC-REF (68.73%), EEG LOC-REF (68.73%), EMG-REF (61.58%), EEG 26-REF (37.45%), EEG 27-REF (36.10%), EEG 28-REF (36.10%), EEG 29-REF (36.10%), EEG 30-REF (36.10%), EEG SP2-REF (24.71%), EEG SP1-REF (24.71%), EEG 32-REF (24.71%), EEG 31-REF (24.71%), EEG C3P-REF (21.43%), EEG C4P-REF (21.43%), EEG LUC-REF (3.28%), EEG EKG-REF (3.28%), EEG RESP2-REF (3.28%), EEG RESP1-REF (3.28%), EEG RLC-REF (3.28%), EEG PG1-REF (0.19%), EEG PG2-REF (0.19%), EEG OZ-REF (0.19%), ECG EKG-REF (0.19%), PULSE RATE (0.19%) |
| *CAPSleep* | **(50 Channel Configurations)** 5-81e3.hdf5, 8-70c4.hdf5, 8-e5e1.hdf5, 8-ed8c.hdf5, 9-1d80.hdf5, 10-a273.hdf5, 11-2037.hdf5, 11-618d.hdf5, 12-9c52.hdf5, 12-476d.hdf5, 12-9ad1.hdf5, 13-f032.hdf5, 14-0b4e.hdf5, 14-b15b.hdf5, 15-04c8.hdf5, 15-b77d.hdf5, 15-eac7.hdf5, 15-4704.hdf5, 16-d600.hdf5, 16-a441.hdf5, 16-66b0.hdf5, 16-83fa.hdf5, 17-7fcb.hdf5, 17-c9e7.hdf5, 18-3b00.hdf5, 18-6721.hdf5, 18-e272.hdf5, 18-1ed6.hdf5, 18-7bbf.hdf5, 18-44bd.hdf5, 18-ec0a.hdf5, 20-bb02.hdf5, 20-83d1.hdf5, 20-1502.hdf5, 21-9aa7.hdf5, 21-be0d.hdf5, 22-3774.hdf5, 22-c70e.hdf5, 22-ead1.hdf5, 22-971d.hdf5, 22-3819.hdf5, 23-83a9.hdf5, 23-7a56.hdf5, 23-4968.hdf5, 23-6bb1.hdf5, 24-8236.hdf5, 24-1436.hdf5, 27-9386.hdf5, 34-a5d0.hdf5, 36-078c.hdf5 | C4-P4 (52.94%), F4-C4 (52.94%), C4-A1 (52.94%), P4-O2 (52.94%), ECG1-ECG2 (51.34%), ROC-LOC (51.34%), EMG1-EMG2 (51.34%), HR (48.13%), SX1-SX2 (47.59%), Fp2-F4 (47.06%), SAO2 (46.52%), DX1-DX2 (45.99%), ECG (43.85%), PLETH (42.25%), EOG E1-M2 (41.18%), EEG C3-M2 (41.18%), EMG chin (41.18%), EEG C4-M1 (41.18%), EEG O2-M1 (41.18%), EEG F4-M1 (41.18%), EOG E2-M2 (41.18%), STAT (40.64%), C3-P3 (37.43%), P3-O1 (37.43%), F3-C3 (37.43%), FP1-F3 (36.90%), F7-T3 (36.36%), F8-T4 (36.36%), T4-T6 (24.06%), T3-T5 (24.06%), TORACE (17.65%), MIC (15.51%), ADDOME (12.83%), Position (7.49%), FP2-F4 (4.28%), Pleth (4.28%), Ox Status (4.28%), ADDDOME (3.21%), LOC (3.21%), ROC (3.21%), T4 (2.67%), C3-A2 (2.67%), P4 (2.14%), O2 (2.14%), O1 (2.14%), F3 (2.14%), F8 (2.14%), P3 (2.14%), T5 (2.14%), T3 (2.14%), FP1 (2.14%), Flusso (2.14%), T6 (2.14%), F7 (2.14%), ECG1 (2.14%), A2 (2.14%), A1 (2.14%), O2-A1 (2.14%), Fp2 (2.14%), EMG2 (2.14%), ECG2 (2.14%), EMG1 (2.14%), TIB Sx (2.14%), F4 (2.14%), TIB Dx (2.14%), C3 (2.14%), C4 (2.14%), ROC-A2 (1.60%), LOC-A1 (1.60%), THE (1.60%), TAG (1.60%), DX2 (1.60%), SX1 (1.60%), SX2 (1.60%), DX1 (1.60%), TERMISTORE (1.60%), Dx1-DX2 (1.60%), C4A1 (1.60%), O1A2 (1.60%), EKG (1.60%), CHIN2 (1.60%), O2A1 (1.60%), CHIN1 (1.60%), F3A2 (1.60%), SpO2 (1.60%), F2-F4 (1.60%), LOC-ROC (1.60%), F4A1 (1.60%), C3A2 (1.60%), milo (1.07%), EMG-EMG (1.07%), EOG-L (1.07%), EOG-R (1.07%), Posizione (1.07%), O1-A2 (0.53%), LOC / A2 (0.53%), ROC / A1 (0.53%), CHIN-0 (0.53%), CHIN-1 (0.53%), F1-F3 (0.53%), EMG (0.53%), abdomen (0.53%), deltoide (0.53%), cannula (0.53%), tib sin (0.53%), ekg (0.53%), toracico (0.53%), Flow (0.53%), Flattening (0.53%), Canula (0.53%), Heart Rate Varia (0.53%), Abdo (0.53%), Torace (0.53%), tib dx (0.53%), EOG sin (0.53%), Tib dx (0.53%), EOG dx (0.53%), Tib sx (0.53%), flow (0.53%), thorax (0.53%), Flusso-0 (0.53%), Flusso-1 (0.53%), Sound (0.53%) |

and heterogeneous electrode configurations, respectively. For datasets with a single configuration, only one file is stored. In contrast, datasets with multiple configurations require multiple files, with each configuration's occurrence proportionally recorded. This dataset design enhances cross-domain generalization and enables principled exploration of electrode heterogeneity in EEG representation learning.

The HEAR Dataset is designed to facilitate heterogeneous training in EEG modeling by preserving the full diversity of electrode configurations while ensuring efficient organization and accessibility. Unlike traditional approaches that select common channels across datasets, our method retains all original channel information and partitions data into subsets based on their respective electrode configurations. Each subset is stored separately in an HDF5 file, enabling flexible heterogeneous processing and channel modeling while maintaining spatial fidelity. This structure not only enhances

Table 15: HEAR Dataset with heterogeneous configurations. (P2)

| Dataset Name | File with Heterogeneous Configurations | Channel Proportions among all Configurations |
|---|---|---|
| *SleepEDFx* | (2 Channel Configurations)
5-851b.hdf5, 7-57a9.hdf5 | EEG Fpz-Cz (100.0%), EEG Pz-Oz (100.0%), EOG horizontal (100.0%), EMG submental (100.0%),
Resp oro-nasal (77.66%), Temp rectal (77.66%), Event marker (77.66%), Marker (22.34%) |
| *CHBMIT* | (12 Channel Configurations)
22-b029.hdf5, 23-2dd3.hdf5, 24-d6eb.hdf5,
24-5e07.hdf5, 25-5929.hdf5, 28-7920.hdf5,
28-6a7b.hdf5, 29-4c37.hdf5, 29-a2cb.hdf5,
29-4c87.hdf5, 31-082d.hdf5, 38-065b.hdf5 | C4-P4 (99.56%), F3-C3 (99.56%), T7-P7 (99.56%), P7-O1 (99.56%), FP1-F3 (99.56%),
P3-O1 (99.56%), C3-P3 (99.56%), FP2-F4 (99.56%), F7-T7 (99.56%), FP1-F7 (99.56%),
F4-C4 (99.56%), CZ-PZ (99.56%), FZ-CZ (99.56%), P8-O2 (99.56%), F8-T8 (99.56%),
FP2-F8 (99.56%), P4-O2 (99.56%), T8-P8-1 (95.48%), T8-P8-0 (95.48%), FT9-FT10 (95.48%),
T7-FT9 (95.48%), P7-T7 (95.48%), FT10-T8 (95.48%), –2 (42.57%), –0 (42.57%),
–1 (42.57%), –3 (42.57%), –4 (38.48%), .-4 (9.33%), .-2 (9.33%), .-3 (9.33%), .-1 (9.33%),
.-0 (9.33%), CP6-Ref (5.83%), FC1-Ref (5.83%), FC5-Ref (5.83%), FC2-Ref (5.83%),
CP1-Ref (5.83%), CP2-Ref (5.83%), CP5-Ref (5.83%), FC6-Ref (5.83%), PZ-OZ (5.69%),
–5 (5.69%), ECG (5.25%), T8-P8 (4.08%), VNS (2.62%), LOC-ROC (1.60%), P3 (0.29%),
F3 (0.29%), C3 (0.29%), CZ (0.29%), FZ (0.29%), PZ (0.29%), FP1 (0.29%), C2 (0.29%),
EKG1-CHIN (0.29%), T8 (0.29%), P8 (0.29%), T7 (0.29%), P7 (0.29%), F7 (0.29%),
O1 (0.29%), O2 (0.29%), F8 (0.29%), FP2 (0.29%), F4 (0.29%), CP4 (0.29%),
CP6 (0.29%), C4 (0.29%), P4 (0.29%), CP2 (0.29%), C6 (0.29%), P7-CS2 (0.15%),
T7-CS2 (0.15%), C3-CS2 (0.15%), P3-CS2 (0.15%), CZ-CS2 (0.15%), PZ-CS2 (0.15%),
FP2-CS2 (0.15%), F7-CS2 (0.15%), C6-CS2 (0.15%), C2-CS2 (0.15%),
O2-CS2 (0.15%), F4-CS2 (0.15%), C4-CS2 (0.15%), P4-CS2 (0.15%), CP6-CS2 (0.15%),
F8-CS2 (0.15%), T8-CS2 (0.15%), P8-CS2 (0.15%), CP4-CS2 (0.15%),
CP2-CS2 (0.15%), LUE-RAE (0.15%), F3-CS2 (0.15%), O1-CS2 (0.15%),
FZ-CS2 (0.15%), FP1-CS2 (0.15%), EKG1-EKG2 (0.15%) |
| *TUAB* | (17 Channel Configurations)
27-3cec.hdf5, 27-2288.hdf5, 28-8134.hdf5,
29-b5ad.hdf5, 29-6d29.hdf5, 30-3158.hdf5,
30-1ba1.hdf5, 30-8cc3.hdf5, 31-8a53.hdf5,
31-11eb.hdf5, 31-372e.hdf5, 33-aec0.hdf5,
34-2300.hdf5, 34-fcd3.hdf5, 35-cc48.hdf5,
35-2d75.hdf5, 36-8e4e.hdf5 | EEG FP1-REF (100.0%), EEG FP2-REF (100.0%), EEG F3-REF (100.0%), EEG F4-REF (100.0%),
EEG C3-REF (100.0%), EEG C4-REF (100.0%), EEG P3-REF (100.0%), EEG P4-REF (100.0%),
EEG O1-REF (100.0%), EEG O2-REF (100.0%), EEG F7-REF (100.0%), EEG F8-REF (100.0%),
EEG T3-REF (100.0%), EEG T4-REF (100.0%), EEG T5-REF (100.0%), EEG T6-REF (100.0%),
EEG A1-REF (100.0%), EEG A2-REF (100.0%), EEG FZ-REF (100.0%), EEG CZ-REF (100.0%),
EEG PZ-REF (100.0%), SUPPR (100.0%), BURSTS (100.0%), IBI (100.0%), EEG T2-REF (99.8994%),
EEG T1-REF (99.8994%), EEG EKG1-REF (99.8994%), PHOTIC-REF (94.7368%),
EEG ROC-REF (93.0607%), EEG LOC-REF (93.0607%), EMG-REF (60.5431%),
EEG 26-REF (55.4811%), EEG 27-REF (54.6430%), EEG 28-REF (54.6430%),
EEG 29-REF (54.6430%), EEG 30-REF (54.6430%), EEG C3P-REF (3.7211%),
EEG C4P-REF (3.7211%), EEG 31-REF (3.7211%), EEG 32-REF (3.7211%), EEG SP1-REF (3.5199%),
EEG SP2-REF (3.5199%), EEG OZ-REF (0.1006%), ECG EKG-REF (0.1006%),
PULSE RATE (0.1006%), EEG PG1-REF (0.0670%), EEG PG2-REF (0.0670%) |
| *TUSL* | (12 Channel Configurations)
27-2288.hdf5, 28-8134.hdf5, 30-8cc3.hdf5,
30-1ba1.hdf5, 32-3a05.hdf5, 33-aec0.hdf5,
33-08ac.hdf5, 33-17fd.hdf5, 34-2300.hdf5,
34-fcd3.hdf5, 36-8e4e.hdf5, 41-7f15.hdf5 | EEG P3-REF (78.57%), EEG P4-REF (78.57%), EEG T2-REF (78.57%), EEG T1-REF (78.57%),
EEG PZ-REF (78.57%), EEG A1-REF (78.57%), EEG CZ-REF (78.57%), EEG FZ-REF (78.57%),
EEG F7-REF (78.57%), EEG C4-REF (78.57%), EEG O1-REF (78.57%), EEG O2-REF (78.57%),
EEG T3-REF (78.57%), EEG F8-REF (78.57%), EEG T4-REF (78.57%), EEG FP1-REF (78.57%),
EEG T6-REF (78.57%), EEG T5-REF (78.57%), EEG A2-REF (78.57%), EEG C3-REF (78.57%),
EEG F4-REF (78.57%), EEG F3-REF (78.57%), EEG FP2-REF (78.57%), EEG EKG1-REF (73.21%),
IBI (73.21%), BURSTS (73.21%), SUPPR (73.21%), PHOTIC-REF (56.25%), EEG 32-REF (38.39%),
EEG 31-REF (38.39%), EEG SP2-REF (35.71%), EEG SP1-REF (35.71%), EEG ROC-REF (33.93%),
EEG LOC-REF (33.93%), EEG C4P-REF (30.36%), EEG C3P-REF (30.36%), EMG-REF (25.89%),
EEG PG2-LE (21.43%), EEG P4-LE (21.43%), EEG O1-LE (21.43%), EEG PG1-LE (21.43%),
EEG OZ-LE (21.43%), EEG CZ-LE (21.43%), EEG PZ-LE (21.43%), EEG EKG-LE (21.43%),
EEG 30-LE (21.43%), EEG FP1-LE (21.43%), PHOTIC PH (21.43%), EEG A1-LE (21.43%),
EEG C4-LE (21.43%), EEG P3-LE (21.43%), EEG A2-LE (21.43%), EEG C3-LE (21.43%),
EEG F4-LE (21.43%), EEG FP2-LE (21.43%), EEG F3-LE (21.43%), EEG O2-LE (21.43%),
EEG F7-LE (21.43%), EEG F8-LE (21.43%), EEG T3-LE (21.43%), EEG T4-LE (21.43%),
EEG T5-LE (21.43%), EEG T6-LE (21.43%), EEG FZ-LE (21.43%), EEG 26-REF (19.64%),
EEG 28-REF (19.64%), EEG 30-REF (19.64%), EEG 29-REF (19.64%), EEG 27-REF (19.64%),
EEG 29-LE (18.75%), EEG 28-LE (18.75%), EEG 27-LE (14.29%), EEG 32-LE (14.29%),
EEG 26-LE (14.29%), EEG 31-LE (14.29%), DC6-DC (14.29%), DC7-DC (14.29%), DC8-DC (14.29%),
DC1-DC (14.29%), DC2-DC (14.29%), DC3-DC (14.29%), DC4-DC (14.29%), DC5-DC (14.29%),
EEG SP1-LE (7.14%), EEG T1-LE (7.14%), EEG SP2-LE (7.14%), EEG T2-LE (7.14%),
EEG EKG-REF (5.36%), EEG RLC-REF (5.36%), EEG LUC-REF (5.36%), EEG RESP2-REF (5.36%),
EEG RESP1-REF (5.36%), EEG RLC-LE (2.68%), EEG LUC-LE (2.68%) |

the model's ability to generalize across various electrode layouts but also fosters adaptability to heterogeneous data environments.

The diverse channel configurations in HEAR provide an ideal training environment for improving model robustness across channels and tasks. By preserving channel variability, our dataset enables models to learn shared representations that transfer effectively between different experimental setups, mitigating the limitations imposed by rigid channel standardization. This approach significantly enhances the model's ability to generalize across multiple tasks, making it particularly well-suited for applications in cross-domain EEG analysis.

Table 16: Excerpt of the global channel dictionary.

| Electrode | System | X | Y | Z |
|-----------|--------|-----|-----|-----|
| Fp1 | 10–20 | -0.0806 | -0.0291 | -0.0413 |
| Cz | 10–20 | -0.0803 | -0.0138 | 0.0292 |
| E21 | EGI 256 | -0.0822 | -0.0475 | -0.0033 |
| E65 | EGI 256 | -0.0811 | -0.0061 | 0.0491 |
| E128 | EGI 256 | 0.0557 | -0.0786 | 0.0566 |
| EEG001 | BioSemi 128 | -0.0806 | -0.0291 | -0.0413 |
| EEG072 | BioSemi 128 | 0.0368 | -0.1008 | 0.0364 |

Furthermore, HEAR serves as a comprehensive EEG feature repository that facilitates multi-task learning by fostering cross-task transfer and feature sharing. The pretraining datasets we constructed are a critical resource for large-scale heterogeneous EEG pretraining, offering a rich and diverse foundation for advancing EEG signal processing. This enables the development of more flexible, transferable, and generalizable EEG models, ultimately paving the way for breakthroughs in brain-computer interfaces, cognitive state classification, and neural decoding across varied experimental paradigms.

## L  HETEROGENEOUS ELECTRODE TRAINING STRATEGY

To enable robust and scalable EEG pretraining across a wide variety of electrode configurations, we designed a heterogeneous training framework that encompasses (1) a unified spatial dictionary, (2) a layout-aware dataloader, (3) a parallel loading strategy for heterogeneous batches, and (4) a distributed synchronization protocol for cross-GPU training. Each component is described below.

**A. Layout-Aware Dataloader**
(Modified via PyTorch)

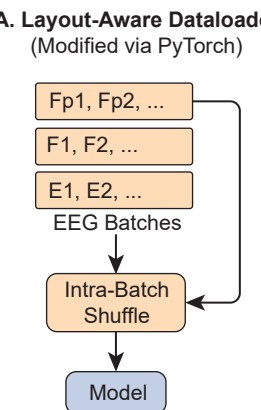

**B. Distributed Inter-GPU Synchronization**

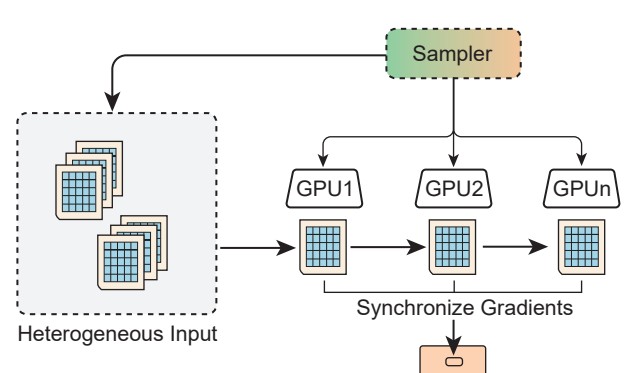

Figure 9: **(A)** Layout-Aware Dataloader organizes EEG samples into batches according to their electrode configuration and applies intra-batch shuffle for diversity. **(B)** Distributed Inter-GPU Synchronization uses a shared sampler to synchronize layout selection across devices, ensuring consistent gradient updates under heterogeneous input.

### L.1  GLOBAL CHANNEL DICTIONARY

We construct a unified global channel dictionary that maps all electrode names across datasets to canonical 3D coordinates. This dictionary enables position-aware modeling for heterogeneous electrode layouts and serves as the basis for coordinate-based embedding in the model. A subset of the dictionary is shown in Table 16, which includes representative electrodes from three commonly used systems: the international 10–20 system, EGI HydroCel 256, and BioSemi 128.

### L.2  LAYOUT-AWARE DATALOADER

To enable robust training over heterogeneous EEG datasets, we design a layout-aware dataloader that groups samples based on their channel configuration. As illustrated in Figure 9A, each EEG sample is assigned to a batch that shares an identical electrode layout (e.g., Fp1–Fp2, F1–F2, or E1–E2). During training, batches are shuffled within each configuration group to ensure intra-layout diversity. Importantly, we modify the PyTorch `Dataset` and `Sampler` modules to register and filter layout-specific metadata. This enables the model to receive input with consistent spatial structure across time, facilitating stable spatial attention and embedding.

### L.3  PARALLEL SUBSET PREFETCHING

To improve I/O efficiency in heterogeneous training, we implement an asynchronous parallel loading strategy: while training is ongoing on the current layout-specific subset, the dataloader launches a background thread to prefetch the next subset. This allows seamless transitions between heterogeneous batches with minimal delay. The system uses PyTorch's `prefetch_factor` and `multiprocessing` support to overlap CPU-GPU data transfer with model computation.

### L.4  DISTRIBUTED INTER-GPU SYNCHRONIZATION

As shown in Figure 9B, we further extend the layout-aware batching mechanism to a multi-GPU setting. A centralized sampler is used to coordinate the selection of layout groups. At each iteration, the sampler broadcasts the current layout index to all GPUs, ensuring that each worker loads batches with the same electrode configuration. Heterogeneous EEG inputs are then distributed across GPUs for parallel processing. During backpropagation, gradients are synchronized across devices via `DistributedDataParallel` (DDP), ensuring consistent model updates. This mechanism maintains architectural and spatial alignment across GPUs, enabling robust representation learning from diverse EEG configurations.

### L.5  PRETRAINING LOSS OF **HEAR**

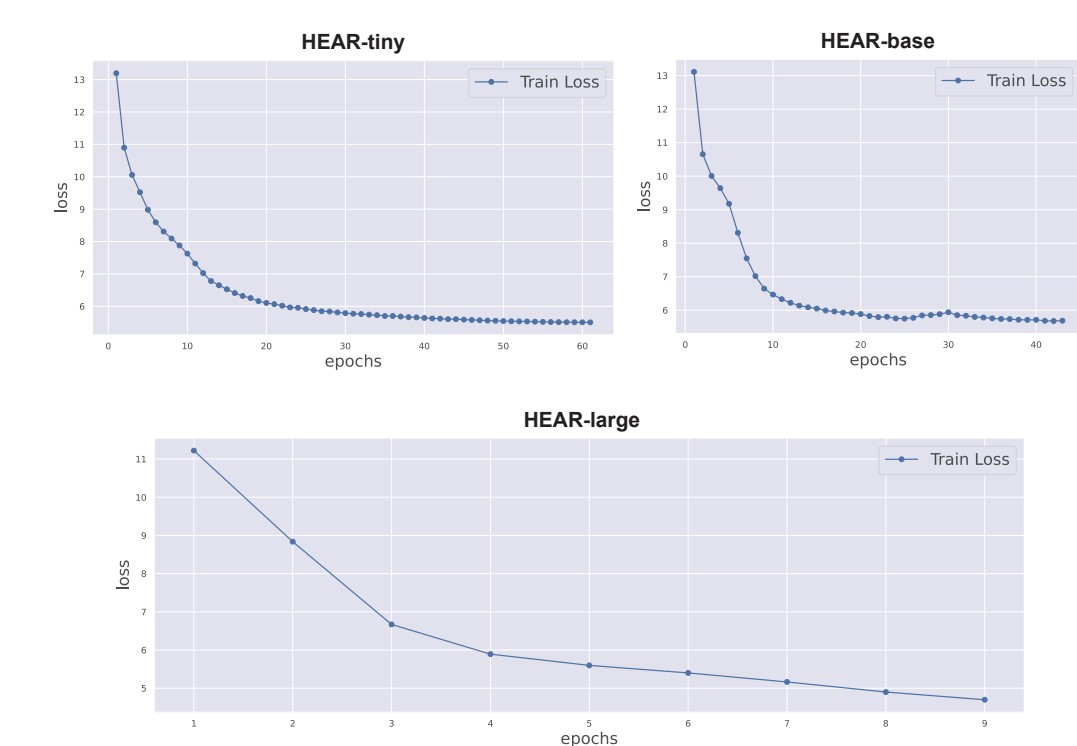

## M LIMITATION

While HEAR demonstrates strong generalization across heterogeneous electrode layouts, several limitations remain. First, our current evaluation focuses on tiny- to base-scale model variants. We have not yet explored the full scaling behavior of EEG foundation models with respect to model capacity and pretraining data volume, leaving open questions about whether larger models or longer training could further improve cross-layout generalization and downstream task performance.

Second, although HEAR supports flexible and previously unseen electrode configurations, it does not explicitly address the integration of derived channels or montage-based systems common in clinical EEG (e.g., bipolar or Laplacian derivations). These systems introduce additional structural dependencies that differ from raw sensor-based input. Extending foundation models to handle such representations in a principled and generalizable manner remains a challenging but important direction for future work.

