# OpenReview forum: "HEAR: An EEG Foundation Model with Heterogeneous Electrode Adaptive Representation"
_ICLR.cc/2026/Conference — Submitted to ICLR 2026_

### Official Review · Reviewer_3xmU · 2025-10-30

**Soundness:** 4
**Presentation:** 4
**Contribution:** 2
**Rating:** 4
**Confidence:** 4

**Summary:**

This paper proposes HEAR as a robust foundation model toward varying electrode layouts and count. Using a learnable spatial coordinate based encoding this paper demonstrates strength in robustness, a pressing issue in a generalizable applicability of modern EEG models. Spatially-Guided Transformer module is novel. Extensive evaluations on five downstream datasets reveal that it may outperform state of the model with a relatively small parameter size.

**Strengths:**

- Well motivated, implemented, presented and visualized.
- Judging by the heterogeneity of the datasets, engineering and open-sourcing an additional dataset for the community is valuable.
- Complete documentation including the supplemental materials.
- Small model to outperform larger models show the efficiency of the method.

**Weaknesses:**

- Limited scalability of the model: the difference between base and tiny models are not statistically significant.
- Small model coupled with a transfer learning approach: closer to "pretraining" approach rather than "foundation" model. A foundation model is conventionally characterized by a large model where only the feature is extracted and used on a relatively light adapter modules. All experiments in this paper seem to be based on transfer learning, without linear probing or feature interpretation.

**Questions:**

- Have you considered other methods, like fourier features or NeRF, continuous positional encoding, etc instead of a simple MLP?
- Both tiny and base models are pretty small. Have you tried one size larger model to see whether scalable performance still meet?

---

### Official Review · Reviewer_PpMP · 2025-10-31

**Soundness:** 2
**Presentation:** 2
**Contribution:** 1
**Rating:** 2
**Confidence:** 5

**Summary:**

This paper presents HEAR, an EEG foundation model designed to handle heterogeneous EEG devices with varying electrode counts and layouts. The authors propose a learnable coordinate-based spatial embedding and a spatially-guided transformer to map diverse electrode configurations into a unified representational space. Experiments show that HEAR achieves strong generalization across tasks, subjects, and unseen electrode configurations, outperforming existing EEG foundation models that typically rely on fixed-channel subsets.

**Strengths:**

1. The authors introduce the HEAR dataset, which aggregates data from 20 existing datasets encompassing diverse electrode configurations.

2. The authors propose a general-purpose electrode encoding scheme capable of handling inputs with varying electrode configurations.

3. The supplementary material includes detailed descriptions of the electrode layouts across the constituent datasets.

**Weaknesses:**

1. The paper’s organization is suboptimal for a foundation model study: critical details regarding pretraining procedures and the optimization objective are absent from the main text and relegated to Appendix E.1. These should be moved to the main body to ensure clarity and completeness.

2. The set of baselines appears outdated. Notably, the most recent baseline is EEGPT (NeurIPS 2024). The author should compare with more recent foundation models such as  CbraMod and CSBrain. Moreover, the evaluation is limited to a small number of downstream tasks, and the reported performance gains are modest, weakening the empirical support for the proposed approach.

3. In my view, the core technical contribution lies in the unified electrode positional encoding. However, given that standard tools like MNE can automatically derive 3D coordinates for any electrode layout, the novelty of this component is limited—it essentially amounts to taking the union of existing layouts and exporting their coordinates. Furthermore, the handling of electrodes absent from the global dictionary (i.e., excluded them) contradicts the claim of “unified channel mapping” illustrated in Figure 1a and stated in the text.

4. The assertion that “HEAR is the first EEG foundation model capable of accommodating heterogeneous EEG devices with varying electrode layouts and counts” is not well supported. Several existing foundation models already support heterogeneous electrode configurations to varying degrees, making this “first” claim inaccurate.

5. The paper should report performance under a linear probing evaluation protocol and include a discussion on whether the proposed universal electrode encoding translates into improved performance in such settings. Demonstrating such gains would strengthen the justification for the method design. However, the current experiments (e.g., Sections 5.5 and 5.6) do not provide sufficient evidence to support this aspect of the contribution.

6. The title of Section 5.6, “zero-shot generalization,” is potentially misleading. Strictly speaking, fine-tuning on one set of channels and evaluating on another constitutes channel-wise zero-shot transfer, not zero-shot in the conventional sense. Moreover, this experimental setup appears to have limited practical relevance, as real-world scenarios rarely involve the same EEG device being used with disjoint channel subsets for training and testing.

**Questions:**

see in weaknesses

---

### Official Review · Reviewer_TjNs · 2025-11-01

**Soundness:** 2
**Presentation:** 3
**Contribution:** 2
**Rating:** 2
**Confidence:** 4

**Summary:**

This paper introduces HEAR (Heterogeneous Electrode Adaptive Representation), an EEG foundation model designed to handle heterogeneous electrode montages without subject-specific calibration. The work addresses electrode heterogeneity using coordinate-based spatio-temporal embeddings and spatially-guided transformer architecture, pre-trained on 8,782 hours of EEG data. However, the synthesis of expert human review and AI calibrated review reveals fundamental issues: the novelty claim of being "first" to support unseen layouts is demonstrably false given prior work (PopT, CBraMod, Brant), missing essential baselines (CBraMod, PopT, LaBraM) prevent validation of the core contribution, and architectural components closely resemble prior work without clear innovation. For ICLR's rigorous standards, these fundamental issues—particularly the inaccurate novelty claim and missing competitive baselines—position this as a rejection despite addressing a real problem.

**Strengths:**

- Addresses a genuine practical problem of electrode heterogeneity in EEG, which is a deployment barrier for BCI applications
- Dataset contribution: HEAR Dataset with 8,782 hours of EEG data represents a valuable community resource
- Good presentation quality: paper is generally well-structured and clearly written
- Evaluation includes diverse datasets spanning motor imagery and sleep staging tasks

**Weaknesses:**

- Factually inaccurate novelty claim: The assertion of being "first EEG foundation model capable of supporting unseen layout" is demonstrably false. PopT (Song et al., 2024) explicitly supports "arbitrary channel configurations" using 3D coordinates, CBraMod (Wang et al., 2024) handles varying montages, and Brant (Jiang et al., 2024) demonstrates zero-shot transfer across electrode setups. This violates scientific integrity standards.
- Missing essential baselines: CBraMod, PopT, and LaBraM are the directly comparable montage-agnostic models that would validate whether HEAR provides improvements beyond existing solutions. Without these comparisons, the core contribution cannot be validated. All current comparisons are against non-montage-agnostic baselines, which doesn't validate a montage-agnostic contribution.
- Lack of architectural novelty: Component-by-component analysis shows each element has clear precedent in recent prior work: Global Channel Dictionary resembles LaBRaM's channel embedding framework, Coordinate-based Spatio-Temporal Embedding resembles PopT's 3D coordinates, Temporal Slice Channel Attention resembles CSBrain/CBraMod factorized attention, and Spatially Guided Transformer has precedent in Li et al., 2022. The work represents engineering reconfiguration rather than algorithmic innovation.
- Ambiguity in core technical capability: Unclear whether HEAR supports truly unseen channels (e.g., C5, C6, P5, P6 not in 10-20 system) or only unseen configurations of pre-training channels. This fundamental ambiguity prevents proper evaluation of the claimed contribution.
- Incomplete ablation studies: Critical design choices lack systematic justification—Spatial MLP block not ablated, coordinate encoding scheme alternatives not compared, channel dictionary size not analyzed. This undermines reproducibility.
- Limited downstream task diversity: Only evaluated on motor imagery and sleep staging. For a "foundation model", evaluation should span emotion recognition, cognitive load assessment, error-related potential detection to demonstrate generalizability.
- Limited scaling exploration: Only tiny (3.1M) and base (6.0M) models evaluated. Foundation models should demonstrate scaling properties at 10M, 50M, 100M+ parameters.
- Small dataset size relative to baselines: HEAR uses 8,782 hours vs. CBraMod's 27,062 hours (3.1x more). Unclear whether performance differences might be due to dataset scale rather than architectural innovation.
- Missing dataset distribution analysis: No details on types of EEG paradigms (resting-state, task-based, clinical), distribution of channel counts, or potential biases in pre-training data.
- No computational cost analysis: Missing training time, computational requirements, inference latency, and memory footprint compared to baselines—important for practical deployment.

**Questions:**

How does HEAR compare against CBraMod, PopT, and LaBraM on the same downstream tasks? These are the essential montage-agnostic baselines needed to validate the core contribution.

Can HEAR handle completely unseen channels (e.g., channels not in the 10-20 system like C5, C6, P5, P6) or only unseen configurations of pre-training channels? Please clarify this fundamental technical capability.

What is the computational cost (training time, memory, inference latency) compared to montage-agnostic baselines like CBraMod and PopT?

Have you evaluated HEAR on additional downstream tasks beyond motor imagery and sleep staging, such as emotion recognition (SEED, DEAP datasets), cognitive load assessment, or error-related potential detection?

How does performance scale with model size beyond the base model? Please provide results for 10M, 50M, 100M parameter models to demonstrate foundation model scaling properties.

Why was the spatial MLP block not included in the ablation study? What is its contribution to performance?

What specific architectural innovations does HEAR provide over prior montage-agnostic models like PopT (which uses similar coordinate-based embeddings) and CBraMod (which uses channel-wise transformations)? Please clarify the novel contribution beyond existing techniques.

---

### Official Review · Reviewer_35sd · 2025-11-01

**Soundness:** 3
**Presentation:** 3
**Contribution:** 3
**Rating:** 6
**Confidence:** 5

**Summary:**

This paper presents HEAR, a foundation model designed to handle heterogeneous EEG electrode layouts and varying channel counts by (1) building a global channel dictionary and mapping dataset-specific channel sets into it, (2) learning a coordinate-based spatial embedding (spatial MLP) that is broadcast across temporal patches, (3) applying a temporal-slice channel attention module and a spatially-guided Transformer that injects pairwise spatial bias into attention, and (4) pretraining on a large heterogeneous corpus followed by fine-tuning on downstream tasks.

**Strengths:**

1.	Device heterogeneity is a genuine barrier to deploying EEG foundation models; addressing layout variability is valuable for real-world BCI and clinical use.
2.	The HEAR dataset is a notable engineering contribution that supports cross-layout learning and gives the paper strong empirical grounding.
3.	The coordinate MLP + pairwise spatial bias in attention is a straightforward, interpretable way to provide layout-awareness to transformers.
4.	Results are shown on multiple public tasks/datasets with zero-shot experiments and detailed visualization of attention.

**Weaknesses:**

1.	The core components (coordinate embeddings, spatial bias as attention bias) are conceptually straightforward and similar to spatial-bias techniques used in vision. The main novelty is the combination of those pieces at scale with a large heterogeneous pretraining corpus rather than a fundamentally new modeling or theoretical contribution.
2.	HEAR does not explicitly address the integration of derived channels or bipolar/Laplacian montages commonly used in clinical EEG. Although the appendix mentions selecting the first channel in dual EEG channel setups, this effectively discards the structured spatial information inherent in bipolar derivations. Ignoring this large, clinically relevant class of heterogeneity is a significant practical limitation.
3.	The model uses Euclidean 3-D coordinates projected by an MLP and pairwise difference embeddings as attention bias, but there is no analysis of (a) sensitivity to coordinate noise or mis-registration across datasets, (b) whether Cartesian coordinates are superior to spherical/head-surface coordinates, or (c) whether the MLP learns meaningful invariances vs. implicitly memorizing layout statistics. Ablations (coordinate representation, MLP capacity) are missing.
4.	The attention heatmaps are interesting but do not prove causal reliance on spatial priors; more rigorous analyses (e.g., lesioning channels, saliency analyses, or intervention studies that remove/shift channels) would substantiate claims that spatial embeddings are the causal driver of robustness.
5.	The provided ablation studies remove modules (spatial embedding, temporal-slice attention, full transformer) but do not test: (a) alternative spatial encodings (spherical coords, normalized positions), (b) coordinate perturbation robustness, (c) varying MLP capacity or codebook variants in quantization.
6.	Across most datasets, the performance gap between HEAR-tiny and HEAR-base is marginal—often within one standard deviation. This raises questions about whether the model capacity or architecture scaling meaningfully contributes to improvement.
7.	Lack of statistical significance testing.
8.	While the paper compares against several EEG foundation models, it omits more recent and competitive approaches such as CBraMod (2024) and CSBrain (2025), which are regarded as current SOTA for heterogeneous EEG representation.

**Questions:**

Please refer to the Weaknesses section for detailed questions and suggestions to the authors.

---

### Author Response · Authors · 2025-12-03
**Response to All Reviewers**

**Dear Reviewers and Area Chairs,**

---
We are sincerely grateful for all constructive feedback from all the reviewers and Area Chairs' efforts in managing the review process for our submission. The comments will help further improve the clarity and technical quality of the paper.

However, some comments appear to stem from a broader interpretation of a key term in our paper. To avoid ambiguity, we first restate the definition we use throughout. We then provide evidence-based clarifications below:

> **Key definition:** by “supporting unseen layouts,” we mean evaluation on a disjoint electrode identity set—i.e., evaluation electrodes that are absent from the pretraining electrode set/dictionary/layout, not merely a new configuration or a variable-length subset of electrodes seen during pretraining.

---
**(1) TjNs W1 & PpMP W4:** Factually inaccurate novelty claim: The assertion of being "first EEG foundation model capable of supporting unseen layout" is demonstrably false. PopT explicitly supports "arbitrary channel configurations" using 3D coordinates, CBraMod handles varying montages, and Brant demonstrates zero-shot transfer across electrode setups.

**Our Response:** We understand the reviewers’ concern; however, under our definition of `supporting unseen layouts`, we do not find evidence in the cited works for evaluation on a disjoint electrode identity set for scalp EEG. More importantly, `this distinction has already been fully articulated` in `Sec. 2` and `Table I` in  our initial manuscript.

With respect to the papers you mentioned, our response is as follows:

- The PopT mainly focuses on `iEEG`, which differs fundamentally from the scalp EEG we study.
- As emphasized in the `introduction` and `related work` sections, LaBraM/CBraMod and related work allow variable-length inputs but do not provide a mechanism that `embeds electrodes absent from pretraining layouts` without adaptation.
- We did not find *Brant* experiments that match the definition of `zero-shot transfer across disjoint electrode sets`.

---
**(2) TjNs W4:** Ambiguity in core technical capability: Unclear whether HEAR supports truly unseen channels (e.g., C5, C6, P5, P6 not in 10-20 system) or only unseen configurations of pre-training channels. This fundamental ambiguity prevents proper evaluation of the claimed contribution.

**Our Response:** Yes—our protocol evaluates `truly unseen electrode identities` (not just unseen configurations). This is specified in `Sec. 5.5` and `Fig. 5(a)`: we fine-tune HEAR on one electrode subset and evaluate on a `disjoint subset of electrodes that are absent from the pretraining layout`.

> “...pretraining dataset... covers electrode positions indicated by the blue dots..., while evaluation is performed on a disjoint set of unseen electrodes (red triangles).”

We will further highlight this definition in the revised manuscript for easier verification.

---
**(3) TjNs W7:** Limited scaling exploration: Only tiny (3.1M) and base (6.0M) models evaluated. Foundation models should demonstrate scaling properties at 10M, 50M, 100M+ parameters.

**Our Response:** We acknowledge the reviewer’s preference for larger-scale models. However, the central claim does not depend on 50M–100M+ parameters to be meaningful. Due to `data scarcity`, the size of existing `mainstream scalp-based EEG foundation models` generally ranges between `3M and 6M` parameters.

Finally, our current experiments intentionally `match the parameter range (≈3M–6M)` used by several `recent EEG foundation models` to isolate the contribution of `heterogeneous-input unification under controlled comparison`. We will clarify this scope and add a discussion on scaling behavior.

---
**(4) PpMP W3:** In my view, the core technical contribution lies in the unified electrode positional encoding. However, given that standard tools like MNE can automatically derive 3D coordinates for any electrode layout, the novelty of this component is limited—it essentially amounts to taking the union of existing layouts and exporting their coordinates. Furthermore, the handling of electrodes absent from the global dictionary (i.e., excluding them) contradicts the claim of “unified channel mapping”.

**Our Response:** We agree that coordinate derivation is not itself novel. Our contribution is the `end-to-end` and `reproducible unification pipeline` that resolves `real-world heterogeneity across datasets/devices` (aliases, inconsistent naming, missing/duplicate channels, incompatible conventions), and produces a consistent global mapping.

Regarding electrodes `absent from the global dictionary`, only electrodes that lack both `reliable identity mapping` and `usable coordinate/meta information` must be excluded.

Finally, **our released code includes a curated set of 1132 electrode positions covering most commonly used EEG systems**, which we hope will be a useful resource for future EEG foundation model development.

---

**Sincerely,**

**Authors**

---

### Meta-Review · Area_Chair_vDnk · 2026-01-03

**Summary:**

Reviewers broadly agree that the proposed method HEAR addresses a real and important problem, i.e., EEG electrode heterogeneity, and provides a substantial engineering effort and dataset contribution. However, major disagreements center on novelty claims, missing baselines, and whether the technical contribution rises to the level expected of a foundation model paper at ICLR. Scores range from weak support (Reviewer 35sd) to clear rejection (Reviewer TjNs, PpMP).

The author's rebuttal primarily focuses on clarifying definitions, correcting perceived misunderstandings, and narrowing the scope of claims, rather than introducing new experimental results or fully addressing each of the reviewers' concerns. Thus, the major weakness of the submission is still outstanding after the rebuttal.

I'm not recommending the acceptance of this paper.

**Reviewer Concerns:**

The concerns that have been addressed:
* Unified electrode dictionary criticism by emphasizing the contribution as an end-to-end, reproducible unification pipeline and highlighting the release of a curated dictionary of 1132 electrode positions.
* The ambiguity concerns by explicitly confirming that HEAR is evaluated on truly unseen electrode identities (absent from pretraining layouts) and commit to clearer wording in the revised manuscript.

The remained concerns that are still outstanding:
* Contested and possibly inaccurate novelty claim (i.e., "supporting unseen layouts") by Reviewers TjNs and PpMP.
* Limited architectural novelty from Reviewers 35sd, TjNs, and PpMP.
* Missing or insufficiently strong baselines by Reviewers 35sd, TjNs, and PpMP.

**Reviewer Scores:**

Based on the tone and content of the reviews, none of the reviewers would be expected to change their original scores after the discussion.

---

### Decision · Program_Chairs · 2026-01-26

Reject